# VidChapters-7M: Video Chapters at Scale

**Antoine Yang**[†], **Arsha Nagrani**[§], **Ivan Laptev**[†], **Josef Sivic**[¶], **Cordelia Schmid**[†]

[†]Inria Paris, DI ENS, CNRS, PSL Research University    [§] VGG, University of Oxford

[¶]Czech Institute of Informatics, Robotics and Cybernetics at the Czech Technical University in Prague

https://antoyang.github.io/vidchapters.html

## Abstract

Segmenting long videos into chapters enables users to quickly navigate to the information of their interest. This important topic has been understudied due to the lack of publicly released datasets. To address this issue, we present VidChapters-7M, a dataset of 817K user-chaptered videos including 7M chapters in total. VidChapters-7M is automatically created from videos online in a scalable manner by scraping user-annotated chapters and hence without any additional manual annotation. We introduce the following three tasks based on this data. First, the video chapter generation task consists of temporally segmenting the video and generating a chapter title for each segment. To further dissect the problem, we also define two variants of this task: video chapter generation given ground-truth boundaries, which requires generating a chapter title given an annotated video segment, and video chapter grounding, which requires temporally localizing a chapter given its annotated title. We benchmark both simple baselines and state-of-the-art video-language models for these three tasks. We also show that pretraining on VidChapters-7M transfers well to dense video captioning tasks in both zero-shot and finetuning settings, largely improving the state of the art on the YouCook2 and ViTT benchmarks. Finally, our experiments reveal that downstream performance scales well with the size of the pretraining dataset. Our dataset, code, and models are publicly available at https://antoyang.github.io/vidchapters.html.

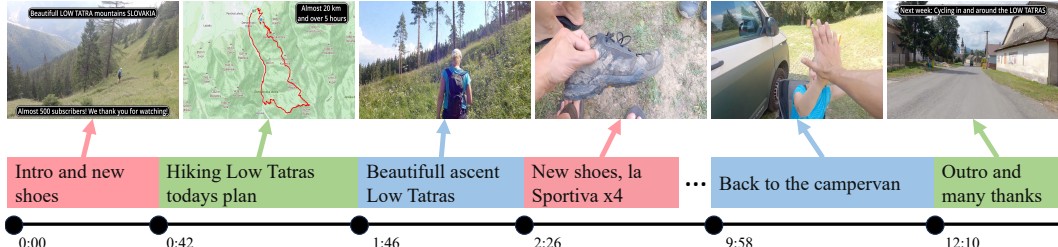

Figure 1: **A video with user-annotated chapters in VidChapters-7M:** the video is temporally segmented into chapters, which are annotated with a chapter title in free-form natural language.

## 1 Introduction

As online media consumption grows, the volume of video content available is increasing rapidly. While searching for specific videos is already a challenging problem, searching within a long video is an even *less* explored task. Manual navigation can often be time consuming, particularly for long videos. A compelling solution for organizing content online is to segment long videos into *chapters* (see Figure 1). Chapters are contiguous, non-overlapping segments, completely partitioning a video. Each chapter is also labeled with a short description of the chapter content, enabling users to quickly

37th Conference on Neural Information Processing Systems (NeurIPS 2023) Track on Datasets and Benchmarks.

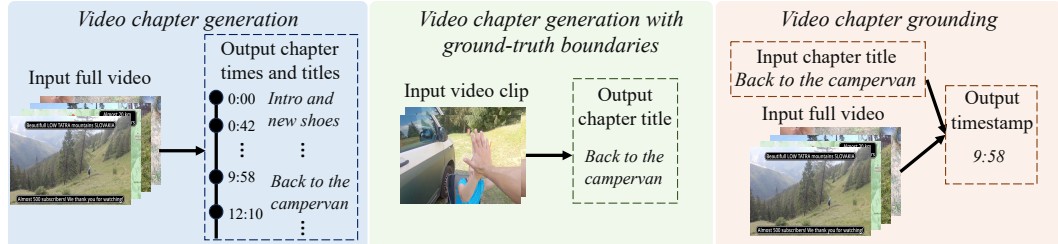

Figure 2: **Illustration of the three tasks defined for VidChapters-7M.**

navigate to areas of interest and easily replay different parts of a video. Chapters also give *structure* to a video, which is useful for long videos that contain inherently listed content, such as listicles [94], instructional videos [62], music compilations and so on.

Given the plethora of content already online, our goal is to explore automatic solutions related to video chaptering - generating chapters automatically, and grounding chapter titles temporally in long videos. While the benefits of automatically chaptering videos are obvious, data for this task is scarce. Video captioning datasets (such as WebVid-10M [5] and VideoCC [64]) consist of short videos (10s in length), and hence are unsuitable. Web datasets consisting of longer videos (HowTo100M [62], YT-Temporal-1B [115]) come with aligned speech transcripts (ASR), which are only weakly related to visual content, and if used as chapter titles would tend to over-segment videos. Moment retrieval [24, 32] or dense video captioning [41, 124] datasets are perhaps the most useful, but do not focus on creating explicit *structure*, and instead describe low-level actions comprehensively. Such datasets are also manually annotated, and hence not scalable and small in size (see Table 1).

To remedy this, we curate VidChapters-7M, a large-scale dataset of user-annotated video chapters automatically scraped from the Web. Our dataset consists of 7M chapters for over 817K videos. Compared to existing datasets, videos in VidChapters-7M are long (23 minutes on average) and contain rich chapter annotations consisting of a starting timestamp and a title per chapter. Our dataset is also diverse, with 12 different video categories having at least 20K videos each, which itself is the size of existing dense video captioning datasets [28, 35, 41, 124]. On top of this dataset we also define 3 video tasks (see Figure 2): (i) *video chapter generation* which requires temporally segmenting the video and generating a chapter title for each segment; (ii) *video chapter generation given ground-truth boundaries* , which requires generating a chapter title given an annotated video segment; and (iii) *video chapter grounding* , which requires temporally localizing a chapter given the chapter title. All three tasks involve parsing and understanding *long* videos, and multi-modal reasoning (video and text), and hence are valuable steps towards story understanding.

For all three tasks, we implement simple baselines as well as recent, state-of-the-art video-text methods [44, 99, 111]. We find that the tasks are far from being solved, demonstrating the value of this problem. Interestingly, we also show that our video chapter generation models trained on VidChapters-7M transfer well to dense video captioning tasks in both zero-shot and finetuning settings, largely improving the state of the art on the YouCook2 [124] and ViTT benchmarks [35]. Moreover, we show that pretraining using both speech transcripts and chapter annotations significantly outperforms the widely used pretraining method based only on speech transcripts [63, 111, 115]. This demonstrates the additional value of our dataset as a generic video-language *pretraining* set. Interestingly, we also find that the transfer performance scales with the size of the chapter dataset.

In summary, our contributions are:

*(i)* We present VidChapters-7M, a large-scale dataset of user-annotated video chapters obtained from the Web consisting of 817K videos and 7M chapters.

*(ii)* Based on this dataset, we evaluate a range of simple baselines and state-of-the-art video-language models on the tasks of video chapter generation with and without ground-truth boundaries, and video chapter grounding.

*(iii)* We show that video chapter generation models trained on VidChapters-7M transfer well to dense video captioning tasks in both zero-shot and finetuning settings, largely improving the state of the art on the YouCook2 [124] and ViTT benchmarks [35], outperforming prior pretraining methods based on narrated videos [111], and showing promising scaling behavior.

Our dataset, code and models are publicly available on our website [1].

| Dataset | Number of videos | Video duration (min) | Number of descriptions | Annotations |
|---|---|---|---|---|
| HowTo100M [62] | 1M | 7 | 136M | Speech transcripts |
| YT-Temporal-1B [115] | **19M** | 6 | **∼ 900M** | Speech transcripts |
| HD-VILA-100M [106] | 3M | 7 | 103M | Speech transcripts |
| ActivityNet Captions [41] | 20K | 3 | 100K | Dense Captions |
| YouCook2 [124] | 2K | 6 | 15K | Dense Captions |
| ViTT [35] | 8K | 4 | 56K | Dense Captions |
| Ego4D [28] | 10K | **23** | 4M | Dense Captions |
| VidChapters-7M (Ours) | 817K | **23** | 7M | **Speech transcripts + User-annotated Chapters** |

Table 1: **Comparison of VidChapters-7M with existing datasets**. We consider open-sourced video datasets that contain dense natural language descriptions aligned over time. VidChapters-7M is much larger than current dense video captioning datasets. Compared to datasets with ASR (top 3 rows), it is smaller in the total number of videos but contains longer videos with richer annotations (chapters).

## 2 Related Work

**Large-scale vision-language datasets.** The development of powerful multi-modal models [3, 15, 23, 34, 36, 37, 45, 47–49, 53, 59, 60, 70, 83, 85, 88, 92, 97, 103, 112, 113, 126] has been made possible by pretraining on large-scale image-caption datasets scraped from the Web such as SBU [66], Conceptual Captions [80], Conceptual-12M [12], LAIT [69], Wikipedia-ImageText [84], RedCaps [18] and LAION-5B [76]. Similarly, many strong video-language models [2, 27, 29, 40, 44, 46, 51, 52, 57, 63, 78, 79, 86, 87, 89, 95, 98, 105, 107–109, 123] have been pretrained on Web-scraped video-text datasets. These datasets are largely composed of short videos paired with captions, e.g. WebVid-10M [5] and VideoCC [64], or narrated videos with speech transcripts aligned over time (ASR), e.g. HowTo100M [62], YT-Temporal-1B [114, 115] and HD-VILA-100M [106]. Our proposed VidChapters-7M dataset is also downloaded from the Web, via a scalable pipeline without the need for expensive manual annotation. Unlike these datasets, VidChapters-7M consists of long videos with user-annotated chapters aligned over time (see Table 1), which significantly differ from ASR (see Section 3.3). Furthermore, most videos in VidChapters-7M *also* contain ASR. Finally, VidChapters-7M is also related to the recent ChapterGen dataset [10], which also consists of user-annotated chapters. However, ChapterGen is several orders of magnitude smaller than VidChapters-7M (10K vs 817K videos) and is not open-sourced at the time of writing.

**Video tasks.** The video chapter generation task requires temporally segmenting the video into chapters, hence is related to video shot detection [74, 75, 82], movie scene segmentation [14, 73], temporal action localization [13, 16, 58, 81, 117, 118] and temporal action segmentation [8, 21, 26, 42, 54, 102]. However, unlike these tasks, video chapter generation also requires generating a free-form natural language chapter title for each segment. Hence this task is also related to video captioning [25, 56, 61, 67, 96, 100, 122], video title generation [4, 116, 120], generic event boundary captioning [101] and dense video captioning [41, 99, 125]. Most related to video chapter generation, the dense video captioning task requires temporally localizing and captioning all events in an untrimmed video. In contrast, video chapter generation requires temporally *segmenting* the video (i.e. the start of the chapter $i + 1$ is the end of chapter $i$, and the chapters cover the full video), and involves generating a chapter title that is substantially shorter than a video caption. We study in more detail the transfer learning between these two tasks in Section 4.4. Finally, the video chapter grounding task is related to temporal language grounding [32, 33, 43, 44, 65, 110, 119, 121]. However, we here focus on localizing a chapter starting point and not a start-end window. Furthermore, most temporal language grounding methods represent the video only with visual inputs, while we also exhibit the benefits of using speech inputs for localizing chapters in videos (see Section 4.3).

## 3 VidChapters-7M: a large-scale dataset of user-chaptered videos

Our goal is to build a large and diverse set of videos annotated with temporarily localized chapter information, consisting of chapter titles and chapter start times. In detail, chapters are contiguous, non-overlapping segments, completely partitioning a video. However manual annotation of chapters is time consuming and expensive and therefore hard to scale. Hence we automatically scrape chapter information from videos available online, as explained in Section 3.1. Then, we perform several processing steps on this data, e.g., to extract speech transcripts, as described in Section 3.2. The

outcome is VidChapters-7M, a dataset of 817K videos with 7M chapter annotations provided by real users online. Finally, we analyze VidChapters-7M in Section 3.3. Details are given next.

## 3.1 Data collection

Since early 2020, YouTube users can create chapters for uploaded videos by annotating them in the YouTube description. The YouTube API, however, currently does not enable explicit search for user-chaptered videos. Hence, our data collection procedure consists of: (i) Collecting a large and diverse set of video candidates (characterized by their 11-character YouTube video ID), which do not necessarily contain user-annotated chapters; (ii) For all video candidates, downloading the video description, automatically selecting videos with user-annotated chapters, extracting video chapters and downloading corresponding videos. We next describe the individual steps in more detail.

**Video candidates.** We start from a large pool of video candidates built from the YT-Temporal-180M dataset [114], which was constructed to be more diverse than prior large video datasets such as HowTo100M [62]. Note that while the released YT-Temporal-180M dataset consists of only 5M videos, the authors collected a larger set of candidates by using YouTube's recommendation algorithm to suggest related videos. We obtained this extended list of 92 million video IDs directly from the authors.

**Extracting chapters from descriptions.** In the description, chapters typically constitute a block with consecutive lines following the format ``<Timestamp>: <Chapter Title>'' or ``<Chapter Title>: <Timestamp>'', where the chapter title is written in free-form natural language and its corresponding start timestamp is written in MM:SS format. The video should contain at least two timestamps listed in ascending order. Hence we download the descriptions for all video candidates and use standard regular expression operations to verify whether a given description contains user-annotated chapters and extract them if so. Note that some videos contain chapters that are automatically generated by YouTube algorithms, however, these generated chapters do not appear in the descriptions and, hence, are excluded by our procedure for data collection. Also note that the video content is only downloaded for user-chaptered videos, which is convenient for both the downloading speed and storage constraints. Finally, we obtain 817K user-chaptered videos, making up 0.9% of all video candidates.

## 3.2 Data processing

We describe below how we process the previously obtained user-chaptered videos to facilitate building efficient video chapter generation models. For reproducibility, we publicly release the resulting speech transcripts and the code for extracting visual features.

**ASR extraction.** We observed that most user-chaptered videos contain speech. Hence, for all videos, we extract speech transcripts aligned in time with the video content (ASR) by applying the Whisper-Large-V2 model [71] on the audio track, using faster-whisper [39] backend for computational efficiency. We found that the Whisper model provides higher-quality ASR compared to the YouTube API ASR service on several data samples from VidChapters-7M. We further use WhisperX [6] to derive accurate word-level timestamps which we use to segment the speech transcript into sentences. For example, the Whisper-Large-V2 model extracts speech segments like *"Right, we're gonna do the Synthetics Dirty Race. No we're not. [...] So we're gonna put two t-shirts and two pairs of jeans in the"* with timestamps 20.478s and 50.465s, and the corresponding first sentence output by WhisperX is *"Right, we're gonna do the Synthetics Dirty Race."* with timestamps 20.538s and 29.26s.

**Visual feature extraction.** Training end-to-end deep learning models from RGB inputs on minutes-long videos is computationally expensive. Hence we extract visual features with CLIP ViT-L/14 backbone [20, 70] at resolution $224 \times 224$ pixels and 1 FPS. This model has been trained to map images to text descriptions with a contrastive loss on 400M Web-scraped image-text pairs.

## 3.3 Data analysis

The result of the previously described pipeline is VidChapters-7M, a dataset of 817,076 user-chaptered videos containing 6,813,732 chapters in total. We randomly split VidChapters-7M in training, validation, and testing splits with 801K, 8.2K, and 8.2K videos, respectively. We analyze VidChapters-

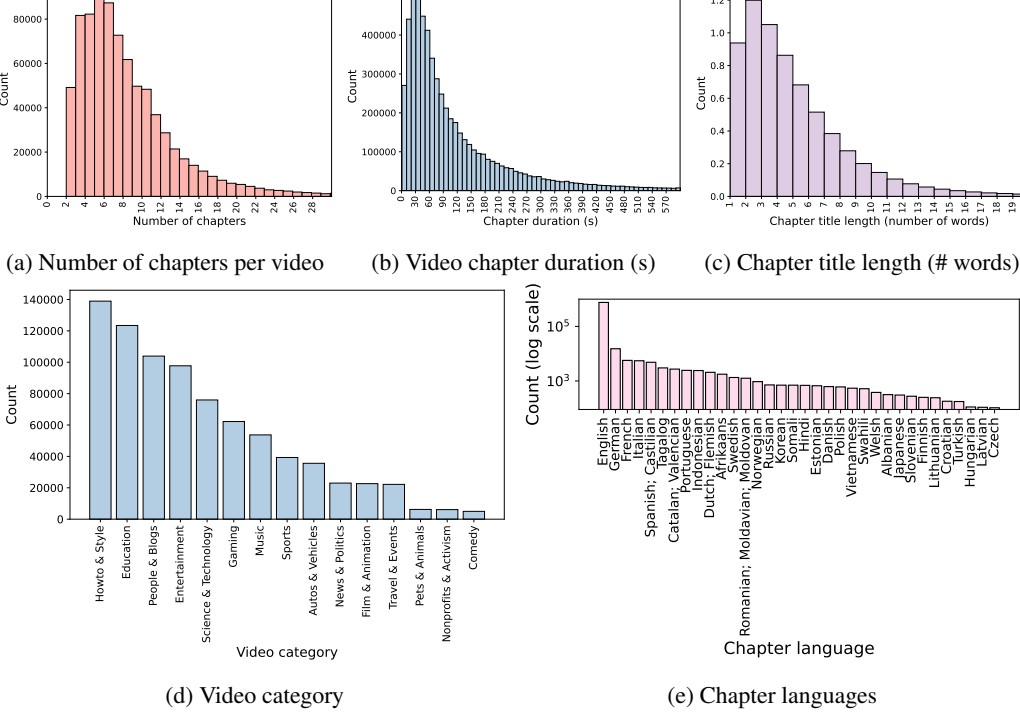

(a) Number of chapters per video  (b) Video chapter duration (s)  (c) Chapter title length (# words)

(d) Video category  (e) Chapter languages

Figure 3: **Statistics of the VidChapters-7M dataset.**

7M below and give examples of annotated videos, more statistics, as well as a datasheet in the Supplementary Material.

**Statistics.** VidChapters-7M is highly diverse and contains 4,894,855 distinct chapter titles. On average, a video contains 8.3 chapters, start times of adjacent chapters are separated by 142.0s seconds, a chapter title contains 5.4 words and a video lasts 1354 seconds. The most represented video category (in YouTube's glossary) is HowTo & Style, making up 17.0% of total videos. The distributions for the number of chapters per video, the video chapter duration, the length of the chapter title, and the video category are illustrated in Figure 3, and further show the diversity of VidChapters-7M, e.g., there are 12 different video categories with at least 20K videos in VidChapters-7M.

**ASR vs Chapters.** 97.3% of videos in VidChapters-7M contain speech transcripts (ASR). However, user-annotated chapters significantly differ from speech transcripts: on average, a video with ASR contains 269.8 speech sentences (vs 8.3 chapter titles), a speech sentence lasts 3.9 seconds (vs 142.0 seconds for chapters) in the video and contains 11.5 words (vs 5.4 words for chapters).

**Biases.** Using the langdetect [17] language detection tool, we find that 92.9%/93.9% of total videos in VidChapters-7M have their chapter titles/ASR in English. However, as shown in Figure 3 (bottom right), the distribution of chapter languages includes a long tail of languages, e.g., 13 languages appear in more than 1K videos of VidChapters-7M. We also use GenBit [77] to measure gender bias in the chapters and ASR. We observe that the percentage of female/male/non-binary gendered words is 19.7%/39.7%/40.7% for the chapters, and 11.6%/35.6%/52.8% for the ASR.

**Ethical considerations.** We employ several techniques to identify harmful visual or language content. We use a classifier [76] built on top of the previously extracted CLIP features to detect not-safe-for-work (NSFW) visual content (such as pornographic and sexualized content). Moreover, we use a language model [30] to detect toxic content in chapter titles and speech transcripts. These processes flag 5,716 (0.70%) visually NSFW videos, 355 (0.04%) videos with toxic chapter titles and 1,368 (0.17%) videos with toxic ASR. We assume the relatively low number of flagged videos is due to the regulations performed by the Web platform used to collect our dataset. Following [76], we refrain from removing these samples to encourage research in fields such as dataset curation and tag them instead. Note that these automated filtering techniques are not perfect and that harmful content may pass.

| Type of chapter titles | Percentage |
|---|---|
| Speech and visual | 49 |
| Audio and visual | 2 |
| Speech-only | 26 |
| Visual-only | 3 |
| Audio-only | 3 |
| Structure-only | 14 |
| Unrelated | 3 |

Table 2: **Manual assessment of the informativeness of chapter titles in the VidChapters-7M dataset over a random sample of 100 videos.** Video chapter titles can be based on speech and vision; audio and vision; vision, audio or speech alone; or only on the structure of the video (*e.g.* "step 1", "step 2" etc). In a small number of cases, video chapters are unrelated to the video content.

**Manual assessment of the quality of annotations.** While chapter titles are manually written and uploaded by real users, sometimes chapter titles are not informative about the content of the video at the corresponding timestamps. To assess the quality of chapter title annotations in our dataset, we inspected a random sample of 100 videos in VidChapters-7M. For each video, we checked if the titles are related to the content of the video chapter and if so which video modalities (ASR, visual or raw audio) they are related to, or if they only refer to the structure of the video (e.g. chapter titles like "step 1", "step 2" etc). Results are presented in Table 2, and show that 83% of videos have chapters related to one or multiple modalities of the video, 14% of videos have chapters only referring to the structure of the video, and 3% of videos have chapters unrelated to the video content.

## 4 Experiments

In this Section, we present the results of models on VidChapters-7M for the full video chapter generation task in Section 4.1, the task of video chapter generation given ground-truth boundaries in Section 4.2 and the video chapter grounding task in Section 4.3. Finally, we study transfer learning from video chapter generation to dense video captioning tasks in Section 4.4.

**Evaluation metrics.** To evaluate the quality of the generated chapter titles (without their positions), we use standard metrics used for visual captioning: BLEU [68] (B), CIDEr [93] (C), METEOR [7] (M) and ROUGE-L [55] (RL). To evaluate video chapter generation as a whole, including the locations of the generated chapters, we follow standard protocols used for dense video captioning, given the similar nature of the two tasks. We use the standard evaluation tool [41] which calculates matched pairs between generated events and the ground truth across IoU thresholds of {0.3, 0.5, 0.7, 0.9}, and compute captioning metrics over the matched pairs. However, these metrics do not take into account the story of the video and give high scores to methods generating many redundant chapters. Hence for an overall evaluation, we also use SODA_c [22] (S) which first tries to find a temporally optimal matching between generated and reference chapters to capture the story of a video, then computes METEOR scores for the matching and derives F-measure scores from the METEOR scores to penalize redundant chapters. To separately evaluate chapter localization, we report the recall (R@Ks, R@K) and the precision (P@Ks, P@K) across various thresholds in terms of the distance to the ground-truth start time or IoU with the ground-truth start-end window. We also report the average recall (R) and average precision (P) across IoU thresholds of {0.3, 0.5, 0.7, 0.9}.

**Implementation details.** Unless stated otherwise, for all models, we use the speech transcripts (ASR) and visual features extracted as explained in Section 3.2. By default, each model is taken from the corresponding official implementation, and all model hyper-parameters are set according to the original papers. We use the Adam optimizer [38] for training and select the final model based on the best validation performance. Our experiments are run on 8 NVIDIA A100 80GB GPUs. More details are included in the Supplementary Material.

### 4.1 Video chapter generation

In this Section, we study the task of video chapter generation that requires temporally segmenting the video and generating a chapter title for each segment.

| Method | Modalities | Pretraining Data | Finetuned | S | B1 | B2 | B3 | B4 | C | M | RL |
|---|---|---|---|---|---|---|---|---|---|---|---|
| Text tiling [31] + Random | Speech | ∅ | ✗ | 0.4 | 0.6 | 0.2 | 0.1 | 0.0 | 0.8 | 0.7 | 0.6 |
| Text tiling [31] + LLaMA [91] | Speech | Text mixture | ✗ | 0.2 | 0.4 | 0.1 | 0.1 | 0.0 | 0.5 | 0.3 | 0.4 |
| Shot detect [90] + BLIP-2 [50] | Visual | 129M image-texts | ✗ | 0.6 | 0.7 | 0.3 | 0.1 | 0.1 | 0.2 | 0.6 | 0.8 |
| Vid2Seq [111] | Speech+Visual | C4 + HowTo100M | ✗ | 0.1 | 0.1 | 0.0 | 0.0 | 0.0 | 0.1 | 0.1 | 0.1 |
| PDVC [99] | Visual | ∅ | ✓ | 6.8 | 9.4 | 3.7 | 1.4 | 0.9 | 35.8 | 9.4 | 11.4 |
| Vid2Seq [111] | Speech | C4 | ✓ | 10.2 | 9.5 | 6.7 | 4.0 | 2.7 | 48.8 | 8.5 | 11.0 |
| Vid2Seq [111] | Speech | C4 + HowTo100M | ✓ | 10.5 | 9.9 | 7.0 | 4.2 | 2.9 | 50.7 | 8.7 | 11.4 |
| Vid2Seq [111] | Visual | C4 | ✓ | 3.1 | 2.3 | 1.5 | 0.6 | 0.5 | 10.9 | 2.2 | 2.9 |
| Vid2Seq [111] | Visual | C4 + HowTo100M | ✓ | 5.5 | 4.5 | 2.8 | 1.2 | 0.9 | 21.1 | 4.1 | 5.5 |
| Vid2Seq [111] | Speech+Visual | C4 | ✓ | 10.6 | 9.9 | 7.0 | 4.2 | 2.8 | 51.3 | 8.8 | 11.6 |
| Vid2Seq [111] | Speech+Visual | C4 + HowTo100M | ✓ | **11.4** | **10.9** | **7.7** | **4.6** | **3.1** | **55.7** | **9.5** | **12.6** |

Table 3: **Video chapter generation (global metrics) on VidChapters-7M test set.** Here, finetuned refers to finetuning on the VidChapters-7M train set, and speech refers to transcribed speech (ASR).

| Method | Modalities | Pretraining Data | Finetuned | R@5s | R@3s | R@0.5 | R@0.7 | P@5s | P@3s | P@0.5 | P@0.7 |
|---|---|---|---|---|---|---|---|---|---|---|---|
| Text tiling [31] | Speech | ∅ | ✗ | 9.4 | 5.8 | 23.6 | 8.9 | 12.6 | 7.9 | 26.0 | 8.8 |
| Shot detect [90] | Visual | ∅ | ✗ | 31.2 | 27.4 | 24.9 | 12.5 | 33.2 | 29.7 | 18.0 | 8.7 |
| Vid2Seq [111] | Speech+Visual | C4 + HowTo100M | ✗ | 10.7 | 9.5 | 5.8 | 0.2 | 23.3 | 18.5 | 1.9 | 0.8 |
| PDVC [99] | Visual | ∅ | ✓ | 21.1 | 17.8 | 31.2 | 22.5 | **45.3** | **40.2** | **47.2** | **26.9** |
| Vid2Seq [111] | Speech | C4 | ✓ | **37.8** | **29.5** | 44.6 | 26.1 | 29.0 | 23.0 | 38.0 | 23.4 |
| Vid2Seq [111] | Speech | C4 + HowTo100M | ✓ | 36.7 | 28.9 | 46.5 | 27.2 | 29.5 | 23.3 | 40.4 | 24.8 |
| Vid2Seq [111] | Visual | C4 | ✓ | 35.3 | 26.4 | 23.6 | 8.7 | 17.9 | 13.6 | 17.2 | 7.1 |
| Vid2Seq [111] | Visual | C4 + HowTo100M | ✓ | 33.5 | 25.0 | 33.0 | 14.5 | 19.5 | 14.7 | 26.2 | 12.5 |
| Vid2Seq [111] | Speech+Visual | C4 | ✓ | 36.3 | 28.6 | 45.8 | 26.9 | 29.9 | 23.8 | 40.9 | 24.9 |
| Vid2Seq [111] | Speech+Visual | C4 + HowTo100M | ✓ | 36.4 | 28.5 | **48.2** | **28.5** | 30.3 | 24.0 | 43.1 | 26.4 |

Table 4: **Video chapter generation (segmentation metrics) on VidChapters-7M test set.**

**Models.** For the video chapter segmentation subtask, we evaluate two zero-shot approaches (i.e., that are not trained on VidChapters-7M): speech text tiling [31], which detects subtopic shifts based on the analysis of lexical co-occurrence patterns, and a visual scene change detection algorithm [90] based on the sum of absolute differences. To derive zero-shot baselines for the full video chapter generation task, we combine text tiling and shot detection with various alternatives that can generate text given text or visual input: a random baseline that predicts a random speech sentence spoken inside the predicted boundaries, LLaMA-7B [91] (prompted to summarize the speech transcript spoken inside the predicted boundaries) and BLIP-2 [50] (prompted to describe the middle video frame of the predicted segment). Finally, we also train and evaluate two state-of-the-art end-to-end dense video captioning models on VidChapters-7M: PDVC [99] which consists of a visual-only DETR-style [11] architecture and Vid2Seq [111] which is a multi-modal sequence-to-sequence model pretrained on the C4 text corpus [72] and on narrated videos with ASR (*e.g.*, YT-Temporal-1B [115]). For Vid2Seq, we also report zero-shot results after pretraining on narrated videos without finetuning on VidChapters-7M.

**Implementation details.** We use the text tiling implementation from the NLTK library [9] which tokenizes the text into pseudosentences of size 50. We use the shot detection software from the FFMPEG library [90] with a confidence threshold of 0.7. For BLIP-2, we use the 3.4B-parameter variant with FLAN-T5-XL [104] and CLIP ViT-L/14 [20, 70]. We reimplement Vid2Seq [111] (originally released in Jax) in PyTorch, use T5-Base pretrained on C4 [72] for initialization and pretrain Vid2Seq on HowTo100M [62]. More details are included in the Supplementary Material.

**Results.** We report the results for video chapter generation using global metrics and localization-only metrics in Tables 3 and 4, respectively. We observe that models trained on VidChapters-7M outperform zero-shot baselines, demonstrating the effectiveness of training on VidChapters-7M. In particular, PDVC [99] has the best precision and Vid2Seq [111] achieves the best results in terms of overall generation and recall. We also find that Vid2Seq's speech-only mode outperforms its visual-only mode and that using both speech and visual inputs leads to the best performance. This demonstrates that video chapter generation is a multi-modal task. Finally, we observe that pretraining using ASR in narrated videos from HowTo100M [62] improves the video chapter generation performance of the Vid2Seq model. Specifically, pretraining on HowTo100M is more beneficial for vision-aware models than for the speech-only model.

| Method | Modalities | Pretraining Data | Finetuned | B1 | B2 | B3 | B4 | C | M | RL |
|---|---|---|---|---|---|---|---|---|---|---|
| Random | Speech | ∅ | ✗ | 2.4 | 1.3 | 0.9 | 0.7 | 10.4 | 2.2 | 4.4 |
| LLaMA [91] | Speech | Text mixture | ✗ | 0.0 | 0.0 | 0.0 | 0.0 | 0.0 | 0.1 | 0.2 |
| BLIP-2 [50] | Visual | 129M image-texts | ✗ | 3.1 | 1.5 | 0.9 | 0.7 | 12.4 | 2.2 | 4.5 |
| Vid2Seq [111] | Speech+Visual | C4 + HowTo100M | ✗ | 2.0 | 1.2 | 0.9 | 0.6 | 0.9 | 0.3 | 0.6 |
| Vid2Seq [111] | Speech | C4 + HowTo100M | ✓ | 21.0 | 15.5 | 12.1 | 10.0 | 105.3 | 11.5 | 24.5 |
| Vid2Seq [111] | Visual | C4 + HowTo100M | ✓ | 10.1 | 5.6 | 3.5 | 2.4 | 47.1 | 5.1 | 14.7 |
| Vid2Seq [111] | Speech+Visual | C4 | ✓ | 21.6 | 15.7 | 12.3 | 10.0 | 110.8 | 11.5 | 26.0 |
| Vid2Seq [111] | Speech+Visual | C4 + HowTo100M | ✓ | **23.5** | **17.2** | **13.4** | **11.0** | **120.5** | **12.6** | **28.3** |

Table 5: **Chapter title generation given ground-truth boundaries on VidChapters-7M test set.**

| Method | Modalities | Pretraining Data | Finetuned | R@10s | R@5s | R@3s | R@1s | R@0.3 | R@0.5 | R@0.7 | R@0.9 |
|---|---|---|---|---|---|---|---|---|---|---|---|
| Random | Speech | ∅ | ✗ | 3.1 | 1.8 | 1.2 | 0.6 | 0.7 | 0.3 | 0.1 | 0.0 |
| BERT [19] | Speech | BookCorpus + Wikipedia | ✗ | 9.0 | 6.8 | 5.4 | 2.9 | 0.6 | 0.3 | 0.1 | 0.0 |
| CLIP [70] | Visual | 400M image-texts | ✗ | 8.1 | 5.2 | 3.7 | 1.4 | 10.7 | 5.2 | 2.3 | 0.5 |
| Moment-DETR [44] | Visual | 5.4K narrated videos [44] | ✗ | 3.2 | 1.6 | 1.1 | 0.5 | 11.3 | 3.6 | 0.8 | 0.1 |
| Moment-DETR [44] | Visual | ∅ | ✓ | **21.8** | **15.5** | **12.4** | **8.3** | **37.4** | **27.3** | **17.6** | **6.4** |

Table 6: **Video chapter grounding on VidChapters-7M test set.**

**Qualitative examples.** See the Supplementary Material.

## 4.2 Video chapter generation given ground-truth boundaries

In this Section, we study the task of generating chapter titles provided correct temporal boundaries of video chapters. This task is a simplification of the previously studied task where we assume perfect temporal segmentation. We adopt the same models and implementation details as previously introduced in Section 4.1.

**Results.** We report results for video chapter generation given ground-truth boundaries in Table 5. Similar to the full video chapter generation task, we observe that solving the task without training on VidChapters-7M is hard. Indeed, LLaMA [91] struggles to summarize the speech content into a chapter title and underperforms the random baseline. Furthermore, BLIP-2 [50] slightly improves over the random baseline. In addition, Vid2Seq [111] in zero-shot mode underperforms the random baseline due to the large domain gap between ASR and chapter titles (see Section 3.3). In comparison, the performance of models trained on VidChapters-7M is significantly higher. Moreover, Vid2Seq's speech-only mode outperforms its visual-only mode, and using both speech and visual inputs is beneficial, confirming the benefit of multi-modal reasoning for the task of generating chapter titles. Finally, pretraining on narrated videos from HowTo100M [62] improves the performance of the Vid2Seq model on VidChapters-7M.

## 4.3 Video chapter grounding

In this Section, we study the task of video chapter grounding that requires a model to temporally localize a chapter start time (or start-end window) given an annotated chapter title (query). Hence, compared to the video chapter generation task, we here assume chapter titles to be given and focus on the temporal chapter localization only.

**Models.** We evaluate three zero-shot alternatives: a random baseline that randomly picks the timestamps of a speech sentence in the video, a BERT [19] baseline that picks the timestamps of the speech sentence that has the closest text embedding with the queried chapter title, and a CLIP [70] baseline picking the frames where the query-frame similarity score drops from the highest scoring frame by a certain threshold $\epsilon$. We also train and evaluate on VidChapters-7M a state-of-the-art end-to-end video grounding model: Moment-DETR [44] which is designed for moment retrieval based on visual inputs. Furthermore, we report zero-shot performance of Moment-DETR obtained with the model checkpoint from Lei et al. [44] pretrained on 5.4K narrated videos with ASR from the QVHighlights dataset [44].

**Implementation details.** We use the `[CLS]` token sequence embedding for the BERT baseline and a threshold of $\epsilon = 0.05$ for the CLIP baseline. More details are provided in the Supplementary Material.

| Method | Modalities | Pretraining Data | YouCook2 (val) | | | | | ViTT (test) | | | | |
|---|---|---|---|---|---|---|---|---|---|---|---|---|
| | | | S | C | M | R | P | S | C | M | R | P |
| PDVC [99] | V | ∅ | 4.4 | 22.7 | 4.7 | — | — | — | — | — | — | — |
| E2ESG [127] | T+V | C4 + WikiHow | — | 25.0 | 3.5 | 20.7 | 20.6 | — | 25.0 | 8.1 | 32.2 | 32.1 |
| Vid2Seq [111] | T+V | C4 + HTM | 8.3 | 48.3 | 9.5 | 27.1 | 27.0 | — | — | — | — | — |
| Vid2Seq [111] | T+V | C4 + YT-Temporal-1B | 7.9 | 47.1 | 9.3 | 27.9 | 27.8 | 13.5 | 43.5 | 8.5 | 42.6 | 46.2 |
| PDVC† | V | ∅ | 4.8 | 28.8 | 5.8 | 22.6 | 33.1 | 9.4 | 40.6 | **16.5** | 19.2 | 37.4 |
| PDVC† | V | VC (Chap.) | 5.9 | 34.7 | 7.5 | 28.8 | **36.4** | 10.1 | 41.5 | 16.1 | 21.3 | 37.2 |
| Vid2Seq† | T+V | C4 + HTM | 8.6 | 53.2 | 10.5 | 29.2 | 26.2 | 14.1 | 44.8 | 8.7 | 43.8 | 44.5 |
| Vid2Seq† | T+V | C4 + VC (ASR+Chap.) | 9.8 | 62.9 | 11.7 | 32.5 | 30.1 | **15.1** | **50.9** | 9.6 | 45.1 | 46.7 |
| Vid2Seq† | T+V | C4 + HTM + VC (ASR) | 8.4 | 50.1 | 10.3 | 29.7 | 26.3 | 14.3 | 45.6 | 8.8 | 43.7 | 44.9 |
| Vid2Seq† | T+V | C4 + HTM + 1% of VC (ASR+Chap) | 8.8 | 52.7 | 10.4 | 29.3 | 27.6 | 13.5 | 41.6 | 8.2 | 44.7 | 42.1 |
| Vid2Seq† | T+V | C4 + HTM + 10% of VC (ASR+Chap.) | 9.9 | 63.9 | 12.1 | 32.4 | 31.4 | 14.5 | 47.4 | 9.2 | 45.3 | 45.9 |
| Vid2Seq† | T+V | C4 + HTM + VC (ASR+Chap.) | **10.3** | **67.2** | **12.3** | **34.0** | 31.2 | 15.0 | 50.0 | 9.5 | **45.5** | **46.9** |

Table 7: **Comparison with the state of the art on the YouCook2 and ViTT dense video captioning benchmarks.** T: Transcribed speech, V: Visual, HTM: HowTo100M [62], VC: VidChapters-7M, Chap.: Chapters. † denote results of our experiments.

| Method | Modalities | Pretraining Data | YouCook2 (val) | | | | | ViTT (test) | | | | |
|---|---|---|---|---|---|---|---|---|---|---|---|---|
| | | | S | C | M | R | P | S | C | M | R | P |
| Text tiling [31] + Random | T | ∅ | 0.3 | 0.9 | 0.3 | 3.8 | 6.6 | 0.3 | 0.6 | 0.6 | 11.6 | 24.4 |
| Text tiling [31] + LLaMA [91] | T | Text mixture | 0.2 | 0.6 | 0.2 | 3.8 | 6.6 | 0.2 | 0.6 | 0.5 | 11.6 | 24.4 |
| Shot detect [90] + BLIP-2 [50] | V | 129M image-texts | 0.6 | 1.0 | 0.5 | 8.9 | 5.5 | 0.2 | 0.1 | 0.2 | 3.1 | 13.7 |
| Vid2Seq [111] | V | C4 + VC (ASR) | 0.0 | 0.0 | 0.0 | 0.0 | 0.0 | 0.0 | 0.0 | 0.0 | 0.2 | 0.8 |
| Vid2Seq [111] | V | C4 + VC (Chap.) | 0.7 | 1.1 | 0.5 | 21.3 | 8.6 | 1.5 | 1.9 | 0.6 | 18.9 | 10.4 |
| Vid2Seq [111] | T+V | C4 + HTM | 0.0 | 0.1 | 0.0 | 0.5 | 0.6 | 0.0 | 0.0 | 0.0 | 0.5 | 1.0 |
| Vid2Seq [111] | T+V | C4 + VC (ASR) | 0.1 | 0.1 | 0.0 | 1.1 | 0.9 | 0.0 | 0.0 | 0.0 | 0.7 | 0.6 |
| Vid2Seq [111] | T+V | C4 + VC (Chap.) | 0.1 | 0.2 | 0.1 | 0.7 | 1.4 | 0.7 | 1.1 | 0.3 | 14.3 | 12.8 |
| Vid2Seq [111] | T+V | C4 + VC (ASR+Chap.) | 3.2 | 10.2 | 2.9 | 20.6 | 19.7 | **9.1** | **30.2** | **6.7** | **33.8** | **40.8** |
| Vid2Seq [111] | T+V | C4 + HTM + VC (ASR) | 0.0 | 0.1 | 0.0 | 1.2 | 0.9 | 0.0 | 0.0 | 0.0 | 0.8 | 0.7 |
| Vid2Seq [111] | T+V | C4 + HTM + 1% of VC (ASR+Chap.) | 2.7 | 7.2 | 2.1 | 18.1 | 17.3 | 5.5 | 15.5 | 4.3 | 31.3 | 37.1 |
| Vid2Seq [111] | T+V | C4 + HTM + 10% of VC (ASR+Chap.) | 3.2 | 11.5 | 3.0 | 19.4 | 19.2 | 6.4 | 21.6 | 5.3 | 31.0 | 38.2 |
| Vid2Seq [111] | T+V | C4 + HTM + VC (ASR+Chap.) | **3.9** | **13.3** | **3.4** | **22.3** | **20.1** | 9.0 | 28.0 | 6.5 | 33.7 | 40.1 |

Table 8: **Zero-shot dense video captioning on the YouCook2 and ViTT benchmarks.** T: Transcribed speech, V: Visual, HTM: HowTo100M [62], VC: VidChapters-7M, Chap.: Chapters.

**Results.** We report results for the video chapter grounding task in Table 6. We first observe that the simple zero-shot baselines based on ASR can decently find start times, but struggle to predict start-end windows due to the important domain gap between ASR and video chapters (see Section 3.3). The CLIP [70] baseline slightly underperforms the BERT baseline [19] at retrieving start times, but is much better at finding start-end windows. Furthermore, the Moment-DETR model [44] trained on VidChapters-7M outperform the zero-shot baselines for both localization of start times and start-end windows, which further demonstrates the effectiveness of training on VidChapters-7M. Finally, we note that Moment-DETR cannot handle speech inputs, but hope that our results showing the benefit of this modality on other tasks in VidChapters-7M will foster research in the localization of language queries in untrimmed videos using multi-modal inputs (vision and speech transcripts).

## 4.4 Transfer learning on dense video captioning

In this Section, we investigate the pretraining of video-language models on our new VidChapters-7M. To this end, we adopt video chapter generation models trained on VidChapters-7M (see Section 4.1) to the tasks of dense video captioning with or without finetuning.

**Datasets.** We use two dense video captioning datasets. **YouCook2** [124] has 2K untrimmed videos of cooking procedures. On average, each video lasts 320s and is annotated with 7.7 temporally-localized sentences. **ViTT** [35] was created to better reflect the distribution of instructional videos in the wild compared to YouCook2, and consists of 8K untrimmed instructional videos. On average, each video lasts 250s and is annotated with 7.1 temporally-localized short tags. For both datasets, we extract speech transcripts and visual features as described in Section 3.2, and follow the standard splits for training, validation and testing. Note that we only use videos available on YouTube at the time of the work, resulting in 10 to 20% less videos than in the original datasets.

**Implementation details.** See Section 4.1 and the Supplementary Material.

**Results after finetuning.** In Table 7, we show that pretraining for video chapter generation on VidChapters-7M greatly improves the downstream dense video captioning performance compared to training from scratch or pretraining only with ASR data as done in previous work [111]. We also find that pretraining both on HowTo100M [62] and VidChapters-7M results in the best overall performance. In particular, the Vid2Seq model pretrained on both HowTo100M and VidChapters-7M largely improves the state of the art on both the YouCook2 and ViTT benchmarks. In detail, on the YouCook2 benchmark, in the setting with C4 + HowTo100M pretraining, we observe that a boost of about 4.9 points in CIDEr is obtained with our reimplementation of Vid2Seq, and that 14.0 additional points in CIDEr are obtained by pretraining on VidChapters-7M. Finally, we report the results of the Vid2Seq model after pretraining on different fractions of VidChapters-7M for a fixed number of iterations. We construct these subsets such that larger subsets include the smaller ones. These results suggest that the scale of the chapter dataset is an important factor in the downstream dense video captioning performance. We conclude that VidChapters-7M opens a promising avenue for multi-modal pretraining. We further show qualitative examples of dense video captioning in the Supplementary Material.

**Zero-shot dense video captioning.** In Table 8, we report results obtained by directly applying video chapter generation models trained on VidChapters-7M for dense video captioning without finetuning for this task. As far as we know, our work is the first to explore this challenging zero-shot setting where no manual annotation of dense video captions is used for training. The Vid2Seq model trained only using ASR data underperforms the random baseline, due to the large domain difference between speech transcripts and dense captions [111]. In the visual-only setting, the variant trained on chapter annotations is better than the variant trained on ASR annotations. In the visual+speech settings, only using chapter annotations does not perform well, as training only on chapters (i.e., without speech) does not enable the model to learn how to use the input speech modality at inference. However, using both ASR and chapter annotations results in a largely better zero-shot dense video captioning performance and outperforms all baselines not trained on VidChapters-7M, demonstrating the complementary nature of the ASR and chapters annotations. Finally, we also observe the benefits of increasing the size of the pretraining dataset of chapters in this setting.

# 5 Conclusion, Limitations, and Societal Impacts

In this work, we presented VidChapters-7M, a large-scale dataset of user-chaptered videos. Furthermore, we evaluated a variety of baselines on the tasks of video chapter generation with and without ground-truth boundaries and video chapter grounding. Finally, we investigated the potential of VidChapters-7M for pretraining video-language models and demonstrated improved performance on the dense video captioning tasks. VidChapters-7M thus provides a new resource to the research community that can be used both as a benchmark for the video chapter generation tasks and as a powerful means for pretraining generic video-language models.

**Limitations.** As it is derived from YT-Temporal-180M [114], VidChapters-7M inherits the biases in the distribution of video categories reflected in this dataset.

**Societal Impacts.** The development of video chapter generation models might facilitate potentially harmful downstream applications, e.g., video surveillance. Moreover, models trained on VidChapters-7M might reflect biases present in videos from YouTube. It is important to keep this in mind when deploying, analysing and building upon these models.

# Acknowledgements

This work was granted access to the HPC resources of IDRIS under the allocation 2023-A0131011670 made by GENCI. The work was funded by Antoine Yang's Google PhD fellowship, the French government under management of Agence Nationale de la Recherche as part of the "Investissements d'avenir" program, reference ANR-19-P3IA-0001 (PRAIRIE 3IA Institute), the Louis Vuitton ENS Chair on Artificial Intelligence, the European Regional Development Fund under project IMPACT (reg. no. CZ.02.1.01/0.0/0.0/15 003/0000468). We thank Jack Hessel and Rémi Lacroix for helping with collecting the dataset, and Antoine Miech for interesting discussions.

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
