# VidChapters-7M: Video Chapters at Scale
# Supplementary Material

**Antoine Yang**[†], **Arsha Nagrani**[§], **Ivan Laptev**[†], **Josef Sivic**[¶], **Cordelia Schmid**[†]

[†]Inria Paris, DI ENS, CNRS, PSL Research University    [§] VGG, University of Oxford

[¶]Czech Institute of Informatics, Robotics and Cybernetics at the Czech Technical University in Prague

https://antoyang.github.io/vidchapters.html

In this Supplementary Material, we present the following items:

*(i)* Additional examples from our VidChapters-7M dataset (Section A).

*(ii)* Qualitative examples of video chapter generation and dense video caption prediction (Section B).

*(iii)* Additional data analysis of our VidChapters-7M dataset (Section C).

*(iv)* Additional implementation details (Section D).

*(v)* Video chapter generation results split by language (Section E).

*(vi)* A datasheet [2] for VidChapters-7M (Section F). Note that in this datasheet, the hosting, licensing, and maintenance plan of VidChapters-7M is presented.

Note that our code, models and the VidChapters-7M dataset can be found on our website [1].

## A    Additional examples from VidChapters-7M

In Figure 1, we provide additional examples that complement Figure 1 of the main paper. These examples illustrate the diversity of the data in VidChapters-7M, e.g., our dataset includes review videos, cooking videos, clothing fitting videos, ASMR videos, and videos of conversations. These examples also show the multi-modal nature of the chapter data. Indeed, chapters depict visual events (e.g., the mini chicken burgers that appear in the second video), conversations (see the last video), or events in the raw audio (e.g., the sound of the crinkly plastic bag in the penultimate video) in various scenarios.

## B    Qualitative examples of video chapter generation and dense video caption prediction

We present qualitative results for video chapter generation and dense video captioning in Figures 2 and 3. Compared with the speech-only model, a key advantage of the speech+visual video chapter generation model is that it can generalize to videos that do not have ASR input, as shown in the first example of Figure 2. Compared with the visual-only variant, the multi-modal model can also benefit from speech cues, as seen in the second example in Figure 2. Moreover, we observe that the dense video captioning model pretrained on VidChapters-7M is more accurate and hallucinates less than the variant not pretrained on VidChapters-7M, see Figure 3.

37th Conference on Neural Information Processing Systems (NeurIPS 2023) Track on Datasets and Benchmarks.

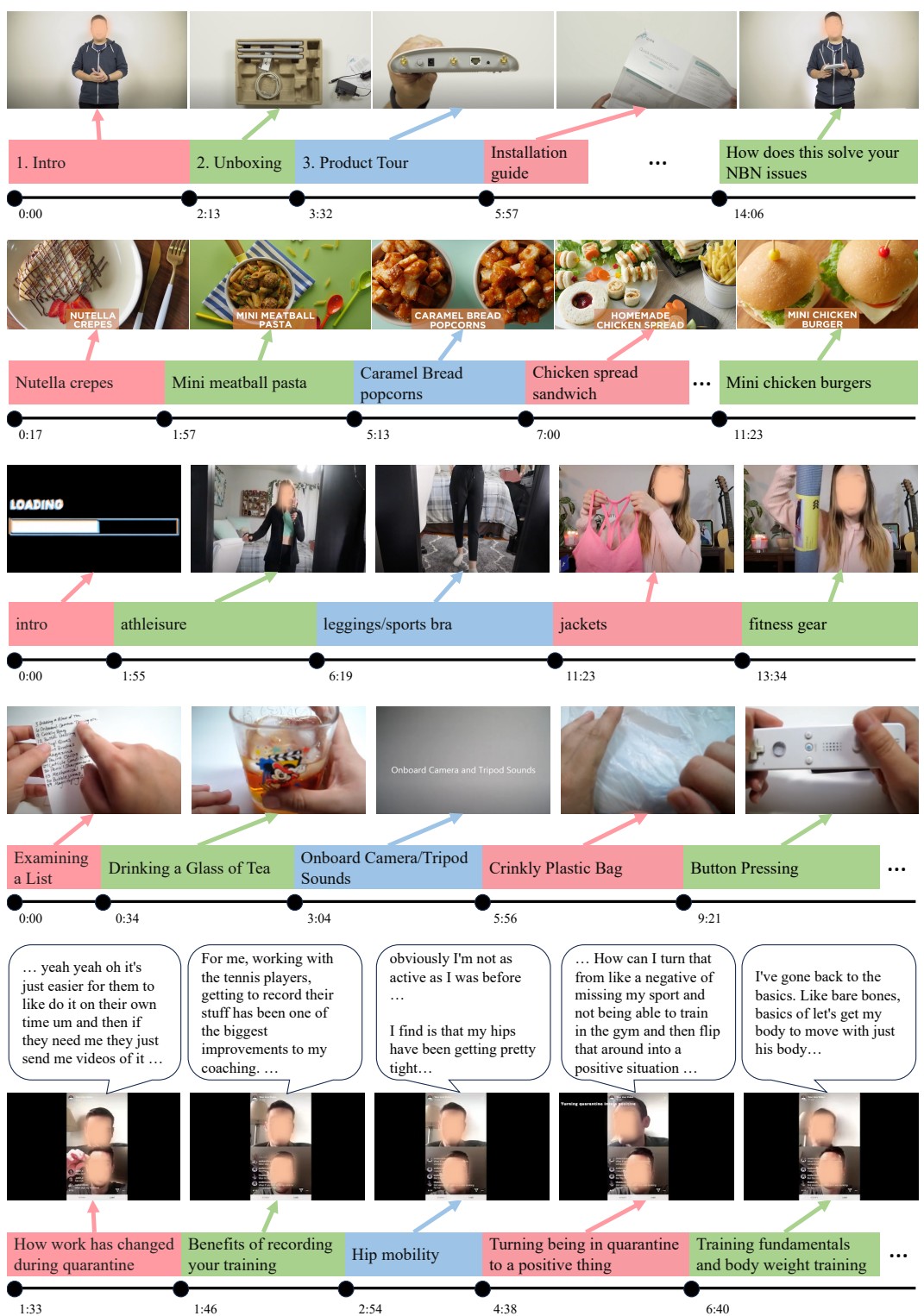

Figure 1: **Additional examples of videos with user-annotated chapters in VidChapters-7M:** Chapters depict visual events (e.g., the mini chicken burgers that appear in the second video), conversations (see the last video), or events in the raw audio (e.g., the sound of the crinkly plastic bag in the penultimate video) in various scenarios.

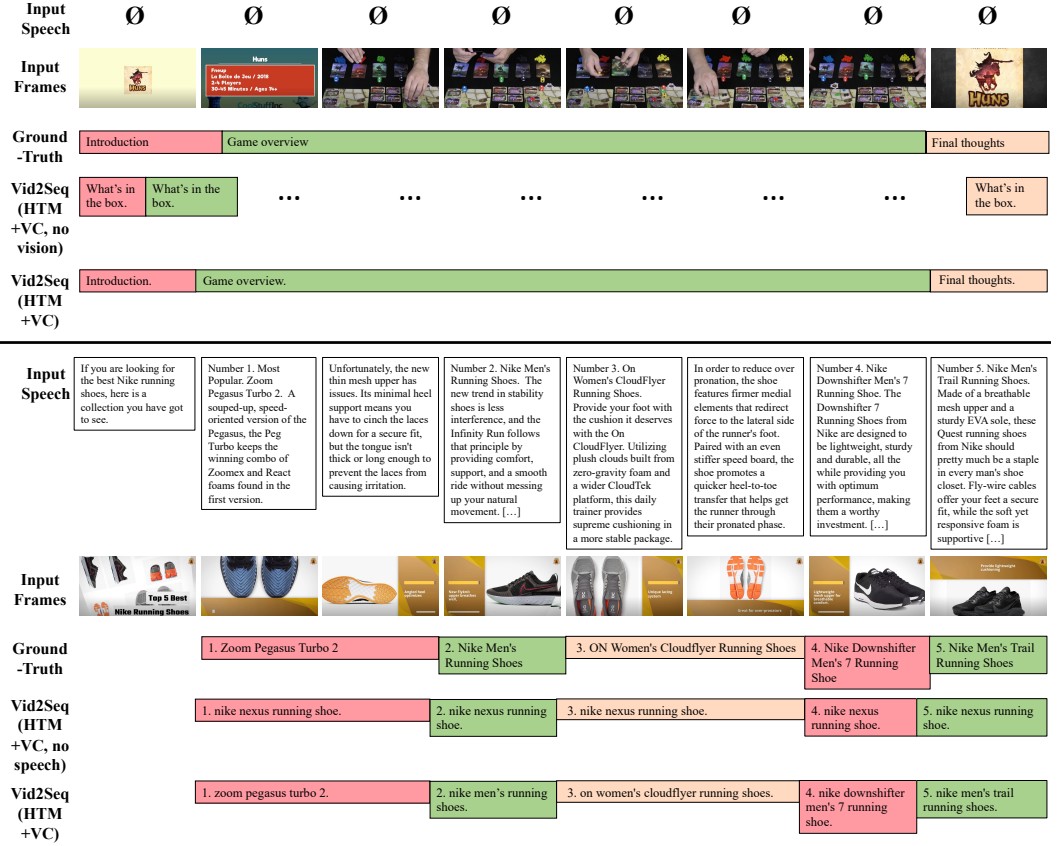

Figure 2: **Examples of video chapter generation using the Vid2Seq model with different input modalities compared with ground-truth on the test set of VidChapters-7M**. The first example shows that the Vid2Seq variant with both speech and visual modalities "Vid2Seq (HTM+VC)" can predict the structure of the input video without the ASR input, unlike the Vid2Seq speech-only variant "Vid2Seq (HTM+VC, no vision)". The second example shows that the Vid2Seq variant with both speech and visual modalities "Vid2Seq (HTM +VC)" can effectively leverage speech cues to detect the names of the depicted and discussed shoes, unlike the Vid2Seq visual-only variant "Vid2Seq (HTM+VC, no speech)".

## C  Additional data analysis of VidChapters-7M

We here complement the analysis of the data in VidChapters-7M provided in Section 3.3 of the main paper. In Figure 4, we show a histogram of the most common chapter titles and word clouds[1] of the chapters titles and ASR content in VidChapters-7M. A few generic chapter titles that outline the structure of the video (e.g., *Intro*, *Introduction*, *Outro*, *Conclusion* and *Start*) appear more than 10K times. Besides, we notice that many videos include chapters about *Unboxing*, *Review*, or *Tips*. This is consistent with the fact that there are many vlogs and 'Howto' videos in VidChapters-7M. We also observe that the most common words in the ASR largely differ from the most common words in the chapter titles, which further shows the difference between these two types of data. To further measure the text-video alignment in the VidChapters-7M dataset, we compute the CLIP cosine similarity between chapter titles and their corresponding video frames and plot the resulting distribution in Figure 5. The average similarity score is 54.6%, and less than 1% of the chapters have a visual-text similarity score below 30%. These statistics demonstrate a good video-text alignment in the VidChapters-7M dataset.

---

[1]To generate the word clouds, we used https://github.com/amueller/word_cloud.

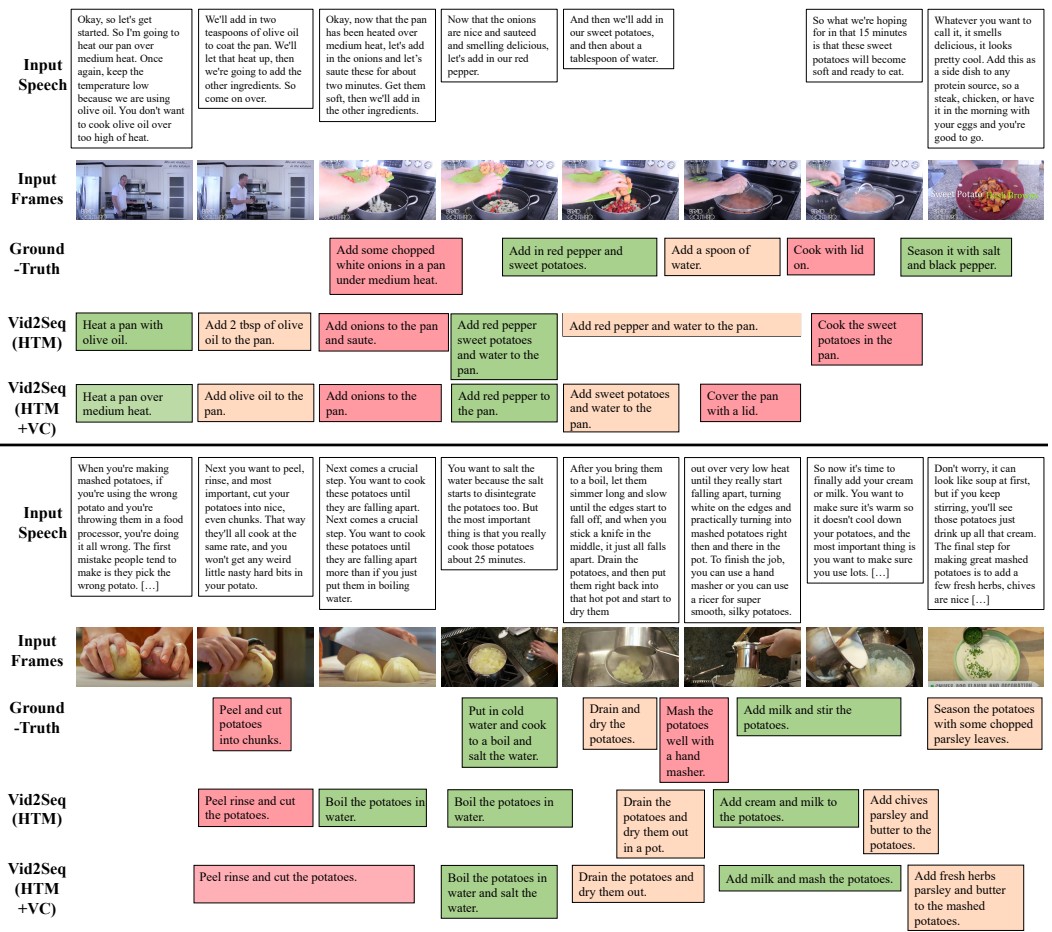

Figure 3: **Examples of dense event captioning of the Vid2Seq model pretrained on VidChapters-7M (vs. not pre-trained), compared with ground-truth, on the validation set of YouCook2.** We find that the model pretrained on VidChapters-7M "Vid2Seq (HTM+VC)" is more accurate and less prone to hallucination. For instance, in the first example (top), the non-VC-pretrained model "Vid2Seq (HTM)" predicts "Add red pepper sweet potatoes and water to the pan." before the sweet potatoes are actually thrown into the pan. In the second example (bottom), the non-VC-pretrained model "Vid2Seq (HTM)" predicts the event "Boil the potatoes in water." twice and predicts the event "Add chives parsley and butter to the potatoes." before it actually happens. The VC-pretrained model "Vid2Seq (HTM+VC)" produces more accurate predictions.

# D    Additional implementation details

In this Section, we present implementation details that complement the information provided in Section 4 of the main paper. We discuss implementation details of our tagging protocol for ethical considerations in Section D.1, models used for video chapter generation and dense video captioning in Section D.2, models used for video chapter generation with ground-truth boundaries in Section D.3, and models used for video chapter grounding in Section D.4.

## D.1    Tagging for ethical considerations

In Section 3.3 of the main paper, we explained how we tag videos for ethical considerations. We here give additional details about this procedure. For the NSFW visual content detector [11], we compute the NSFW score at every frame (at 1 FPS) and tag videos with an average score above 0.5. For the toxic content detection model [3], we compute the toxicity score at every chapter title / ASR sentence and tag videos where the chapter titles / ASR have an average toxicity score above 0.5.

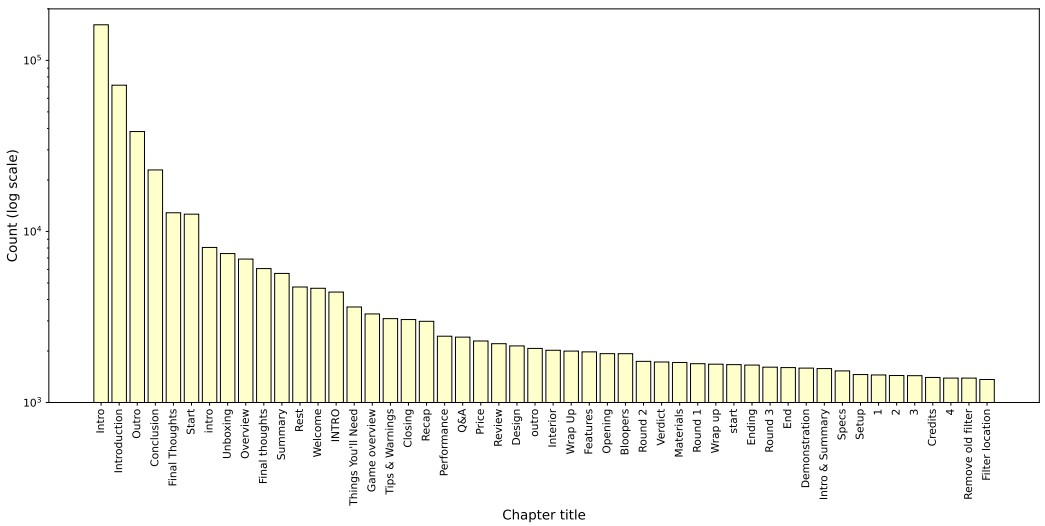

(a) Most common chapter titles

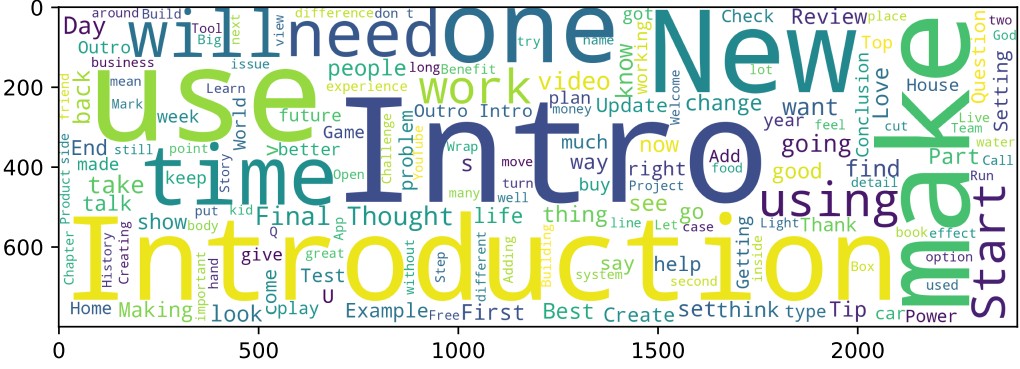

(b) Word clouds of chapter titles.

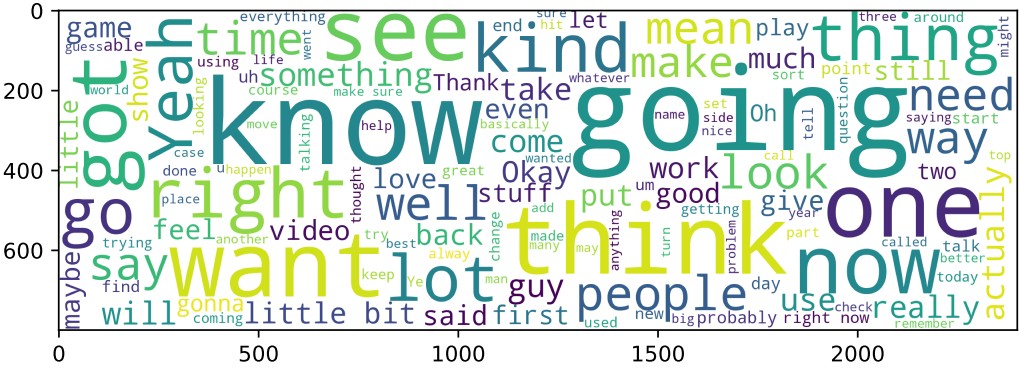

(c) Word clouds of ASR.

Figure 4: **Additional statistics of the VidChapters-7M dataset.**

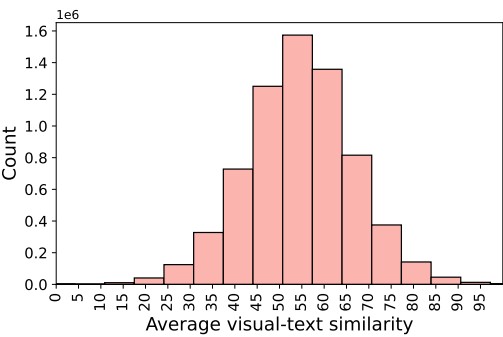

Figure 5: **Average visual-text similarity between chapter titles and the corresponding video frames as measured by CLIP cosine similarity (rescaled between 0 and 100) in VidChapters-7M.**

## D.2 Video chapter generation and dense video captioning

**LLaMA [13].** We use the following prompt: `Summarize the following speech transcript in a chapter title. Transcript: <ASR> Chapter title:` where the ASR is the concatenation of all speech sentences spoken during a given video segment.

**BLIP-2 [6].** We use the following prompt: `Summarize the image in a chapter title. Chapter title:`, and use the middle frame of the predicted video segment.

**PDVC [14].** We use PDVC's official codebase. PDVC includes a caption decoder that relies on dataset-specific word vocabularies. To adapt PDVC to VidChapters-7M/YouCook2/ViTT, we construct a vocabulary made with all words that appear at least 50/2/3 times in the dataset (33,598/3,815/1,607 words). For transfer learning from VidChapters-7M to YouCook2/ViTT, we initialize the downstream dataset-specific word embedding layer with the weights of the corresponding word embedding in the pretrained model. We subsample or pad the sequence of frames to 100 frames. For all datasets, we use 100 queries and train with a constant learning rate of $5e^{-5}$, weight decay $1e^{-4}$ and batch size 1 on an NVIDIA V100 32GB (as the official codebase is not compatible with higher batch sizes or multi-gpu training) . We train on VidChapters-7M/YouCook2/ViTT for 5/30/30 epochs. The training on VidChapters-7M lasts about a week.

**Vid2Seq [16].** We reimplement Vid2Seq (originally released in Jax) in PyTorch. For initialization, we use the T5-Base language model pretrained on the C4 text corpus [10]. Vid2Seq is originally pretrained on YT-Temporal-1B [18] using a generative and denoising objective in the speech sequence. Due to computational limitations, we instead pretrain Vid2Seq on the smaller HowTo100M dataset [8] with the same objectives. Then we train Vid2Seq on VidChapters-7M with the next token prediction objective in the chapter sequence and the denoising objective in the speech sequence. Finetuning on YouCook2/ViTT is done with the next token prediction objective in the dense video captioning sequence and the denoising objective in the speech sequence. We subsample or zero-pad the sequence of frames to 100 frames. The text encoder and decoder sequence are truncated or padded to 1000 and 256 tokens, respectively. For all datasets, we use a learning rate of $3e^{-4}$ warmed up linearly (from 0) for the first 10% of iterations and following a cosine decay (down to 0) for the remaining 90%. We train for 6/10/40/20 epochs on HowTo100M/VidChapters-7M/YouCook2/ViTT. We use a batch size of 64 videos split on 8 NVIDIA A100 80GB for HowTo100M/VidChapters-7M, and 16 videos split on 8 NVIDIA V100 32GB for YouCook2/ViTT. The training on HowTo100M or VidChapters-7M takes about 2 days.

## D.3 Video chapter generation with ground-truth boundaries

**LLaMA [13] and BLIP-2 [6].** See Section D.2.

**Vid2Seq [16].** To adapt the model pretrained on HowTo100M (see Section D.2) to video chapter generation with ground-truth boundaries, we remove the model weights corresponding to the time tokens (in the token embedding layers and the token prediction layer). We train for 20 epochs on VidChapters-7M using the next token prediction objective in the sequence of tokens corresponding to a single chapter title. We construct training batches by sampling a chapter title with its associated

| Method | Modalities | Pretraining Data | Finetuned | S | B1 | B2 | B3 | B4 | C | M | RL |
|---|---|---|---|---|---|---|---|---|---|---|---|
| Text tiling [4] + Random | Speech | ∅ | ✗ | 0.5 | 0.8 | 0.2 | 0.1 | 0.0 | 0.9 | 0.8 | 0.7 |
| Text tiling [4] + LLaMA [13] | Speech | Text mixture | ✗ | 0.3 | 0.5 | 0.2 | 0.1 | 0.0 | 0.5 | 0.4 | 0.4 |
| Shot detect [12] + BLIP-2 [6] | Visual | 129M image-texts | ✗ | 1.3 | 1.5 | 0.7 | 0.4 | 0.2 | 4.7 | 1.4 | 1.6 |
| PDVC [14] | Visual | 129M image-texts | ✓ | 6.6 | 9.0 | 3.8 | 1.5 | 1.0 | 36.0 | 9.1 | 11.0 |
| Vid2Seq [16] | Speech+Visual | C4 | ✓ | 10.8 | 10.3 | 7.6 | 4.9 | 3.4 | 54.8 | 9.1 | 11.9 |
| Vid2Seq [16] w/ mT5 | Speech+Visual | mC4 | ✓ | 10.4 | 9.9 | 7.2 | 4.7 | 3.3 | 52.0 | 8.7 | 11.3 |
| Vid2Seq [16] | Speech+Visual | C4 + HowTo100M | ✓ | **11.5** | **11.1** | **8.1** | **5.1** | **3.6** | **58.8** | **9.7** | **12.8** |

Table 1: **Video chapter generation (global metrics) on the VidChapters-7M test set restricted to videos with English chapter titles and ASR.** Here, finetuned refers to finetuning on the VidChapters-7M train set, and speech refers to transcribed speech (ASR).

| Method | Modalities | Pretraining Data | Finetuned | S | B1 | B2 | B3 | B4 | C | M | RL |
|---|---|---|---|---|---|---|---|---|---|---|---|
| Text tiling [4] + Random | Speech | ∅ | ✗ | 0.6 | 1.7 | 1.3 | 1.3 | 1.1 | 12.8 | 1.5 | 1.6 |
| Text tiling [4] + LLaMA [13] | Speech | Text mixture | ✗ | 0.1 | 0.3 | 0.2 | 0.0 | 0.0 | 0.0 | 0.2 | 0.2 |
| Shot detect [12] + BLIP-2 [6] | Visual | 129M image-texts | ✗ | 0.6 | 0.4 | 0.2 | 0.0 | 0.0 | 1.3 | 0.6 | 0.5 |
| PDVC [14] | Visual | 129M image-texts | ✓ | 5.4 | **11.6** | 0.0 | 0.0 | 0.0 | 29.4 | **12.4** | **14.9** |
| Vid2Seq [16] | Speech+Visual | C4 | ✓ | 9.1 | 8.4 | 5.2 | 1.0 | 0.9 | 34.1 | 6.1 | 10.1 |
| Vid2Seq [16] w/ mT5 | Speech+Visual | mC4 | ✓ | 8.8 | 8.1 | 5.9 | 1.7 | 1.8 | 38.4 | 6.1 | 10.1 |
| Vid2Seq [16] | Speech+Visual | C4 + HowTo100M | ✓ | **10.9** | 9.6 | **5.4** | **1.7** | **1.7** | **43.2** | 8.1 | 8.1 |

Table 2: **Video chapter generation (global metrics) on the VidChapters-7M test set restricted to videos with German chapter titles and ASR.** Here, finetuned refers to finetuning on the VidChapters-7M train set, and speech refers to transcribed speech (ASR).

video clip at each iteration (i.e., an epoch corresponds to seeing one chapter title for all videos). The text encoder and decoder sequence are truncated or padded to 256 and 32 tokens, respectively. We use a learning rate of $3e^{-4}$ warmed up linearly (from 0) for the first 10% of iterations and following a cosine decay (down to 0) for the remaining 90%. We use a batch size of 512 videos split on 8 NVIDIA A100 80GB for VidChapters-7M. The training takes about a day.

### D.4  Video chapter grounding

**Moment-DETR [5].** We use Moment-DETR's official codebase. We train with the AdamW optimizer [7], a constant learning rate of $3e^{-4}$, and a batch size of 256 videos split on 8 NVIDIA A100 80GB. We use a FPS of 1/3 and subsample or zero-pad the sequence of frames to 1200 frames. We use a maximum number of text query tokens of 77. We train for 50 epochs on VidChapters-7M, where an epoch corresponds to seeing one chapter title for all videos, which takes about 2 days.

## E  Video chapter generation results split by language

We report video chapter generation results on the VidChapters-7M dataset split by language for both English and German in Tables 1 and 2, respectively. We find that training on VidChapters-7M is beneficial for both languages. Interestingly, pretraining on HowTo100M (which is a dataset in English) improves results on English as well as German. We also observe that the quantitative results in German are lower than in English. Finally, we report results of the Vid2Seq model with the multi-lingual language model mT5 [15] pretrained on the multi-lingual dataset mC4 [15]. We find that this variant performs a bit worse on English but slightly better on German compared to the Vid2Seq variant based on T5 pretrained on the C4 corpus.

## F  Datasheet for VidChapters-7M

Datasheets for datasets introduced by Gebru et al. [2] serve as a medium of communication between the creators and users of a dataset. They effectively consolidate the motivation, creation process, composition, and intended uses of a dataset as a series of questions and answers. In this Section, we provide a datasheet for the VidChapters-7M dataset.

## Motivation

Q1. **For what purpose was the dataset created?** *Was there a specific task in mind? Was there a specific gap that needed to be filled? Please provide a description.*

- The VidChapters-7M dataset was created to explore the task of video chapter generation, which enables users to quickly navigate to the information of their interest.

Q2. **Who created this dataset (e.g., which team, research group) and on behalf of which entity (e.g., company, institution, organization)?**

- Five researchers have created VidChapters-7M: Antoine Yang (Inria and DI ENS), Arsha Nagrani (VGG, University of Oxford), Ivan Laptev (Inria and DI ENS), Josef Sivic (CIIRC CTU) and Cordelia Schmid (Inria and DI ENS).

Q3. **Who funded the creation of the dataset?** *If there is an associated grant, please provide the name of the grantor and the grant name and number.*

- We collected VidChapters-7M without any monetary costs, since no part of our dataset requires annotations from crowd workers or contractors. This research work was funded by Antoine Yang's Google PhD fellowship, the French government under management of Agence Nationale de la Recherche as part of the "Investissements d'avenir" program, reference ANR-19-P3IA-0001 (PRAIRIE 3IA Institute), the Louis Vuitton ENS Chair on Artificial Intelligence, the European Regional Development Fund under project IMPACT (reg. no. CZ.02.1.01/0.0/0.0/15 003/0000468). However, note that this article solely reflects the opinions and conclusions of its authors and not of its funders.

Q4. **Any other comments?**

- No.

## Composition

Q5. **What do the instances that comprise the dataset represent (e.g., documents, photos, people, countries)?** *Are there multiple types of instances (e.g., movies, users, and ratings; people and interactions between them; nodes and edges)? Please provide a description.*

- Each instance in VidChapters-7M represents a YouTube video.

Q6. **How many instances are there in total (of each type, if appropriate)?**

- There are 817K instances in VidChapters-7M.

Q7. **Does the dataset contain all possible instances or is it a sample (not necessarily random) of instances from a larger set?** *If the dataset is a sample, then what is the larger set? Is the sample representative of the larger set (e.g., geographic coverage)? If so, please describe how this representativeness was validated/verified. If it is not representative of the larger set, please describe why not (e.g., to cover a more diverse range of instances, because instances were withheld or unavailable).*

- VidChapters-7M is a small sample drawn from all the data uploaded to YouTube. Millions of videos are uploaded on YouTube every day. We started from a subset of 92 million YouTube video candidates, which consists of videos recommended in videos from the YT-Temporal-180M dataset [17]. We selected the videos from this subset (817K instances) that contain user-annotated chapters. Hence, VidChapters-7M data does not fully represent YouTube.

Q8. **What data does each instance consist of? "Raw" data (e.g., unprocessed text or images) or features?** *In either case, please provide a description.*

- Each instance in VidChapters-7M consists of four metadata fields:
  - `"video_id"`: Unique alphanumeric ID of the video (assigned by YouTube).
  - `"url"`: Static URL for downloading the video, e.g., `https://www.youtube.com/watch?v=<video_id>`.
  - `"asr"`: ASR transcripts aligned over time.
  - `"chapters"`: Chapters aligned over time.

Q9. **Is there a label or target associated with each instance?** *If so, please provide a description.*

– We use the chapters as labels in this work, though it might be also possible to use auxiliary information (like video titles or tags).

Q10. **Is any information missing from individual instances?** *If so, please provide a description, explaining why this information is missing (e.g., because it was unavailable). This does not include intentionally removed information, but might include, e.g., redacted text.*

– No and yes. No, because all the metadata fields for every instance are filled with valid values. Yes, because the "url" for some instances may not retrieve the underlying video. This may happen if the YouTube user (author) removes the video from YouTube. Such deletions reduce our dataset size over time, however, video deletions are rare.

Q11. **Are relationships between individual instances made explicit (e.g., users' movie ratings, social network links)?** *If so, please describe how these relationships are made explicit.*

– Relationships between individual instances (e.g., videos made by the same creator) are not made explicit in our work, though this is a possibility for future work.

Q12. **Are there recommended data splits (e.g., training, development/validation, testing)?** *If so, please provide a description of these splits, explaining the rationale behind them.*

– We randomly split our data into training, validation, and testing sets. The training, validation, and testing sets are meant for training, development, and evaluation, respectively.

Q13. **Are there any errors, sources of noise, or redundancies in the dataset?** *If so, please provide a description.*

– VidChapters-7M is inherently noisy since YouTubers are free to write the chapters that they want.

Q14. **Is the dataset self-contained, or does it link to or otherwise rely on external resources (e.g., websites, tweets, other datasets)?** *If it links to or relies on external resources,*
   *(a) Are there guarantees that they will exist, and remain constant, over time?*
   *(b) Are there official archival versions of the complete dataset (i.e., including the external resources as they existed at the time the dataset was created)?*
   *(c) Are there any restrictions (e.g., licenses, fees) associated with any of the external resources that might apply to a future user? Please provide descriptions of all external resources and any restrictions associated with them, as well as links or other access points, as appropriate.*

– We do not distribute videos of our dataset to respect YouTube user privacy and to limit our storage budget. Instead, we provide video URLs ("url", Q8) that point to videos hosted on YouTube servers. In response to sub-questions:
   (a) These video servers ensure stable access unless the YouTube user deletes their video.
   (b) Yes, YouTube archives all the metadata of submitted videos. For videos, YouTube only archives the URL and not the media content, giving full control of accessibility to the users.
   (c) All video URLs are freely accessible. It is unlikely for video servers to restrict access in the future, given their free accessibility over the past decade.

Q15. **Does the dataset contain data that might be considered confidential (e.g., data that is protected by legal privilege or by doctor-patient confidentiality, data that includes the content of individuals non-public communications)?** *If so, please provide a description.*

– No, the videos included in VidChapters-7M do not cover topics that may be considered confidential. All videos were publicly shared on YouTube prior to inclusion in VidChapters-7M.

Q16. **Does the dataset contain data that, if viewed directly, might be offensive, insulting, threatening, or might otherwise cause anxiety?** *If so, please describe why.*

– The scale of VidChapters-7M means that we are unable to manually verify the contents of all videos and chapters. However, YouTube removes videos that contain offensive content or do not follow their community guidelines. Furthermore, we employed additional mitigation techniques on VidChapters-7M:
   (a) We tagged all instances whose video frames were predicted as NSFW by an off-the-shelf detector [11].
   (b) We tagged all instances whose chapter titles or speech transcripts were predicted as toxic by a language model [3].

Q17. **Does the dataset relate to people?** *If not, you may skip remaining questions in this section.*

– The dataset pertains to people in that people upload videos to YouTube and write descriptions that include chapter annotations. Furthermore, most videos in VidChapters-7M have people speaking and/or appearing.

Q18. **Does the dataset identify any subpopulations (e.g., by age, gender)?** *If so, please describe how these subpopulations are identified and provide a description of their respective distributions within the dataset.*

– VidChapters-7M does not explicitly identify any subpopulations. Since most videos contain people and chapters are free-form natural language written by YouTube users, it is possible that some chapters may identify people appearing in individual videos as part of a subpopulation.

Q19. **Is it possible to identify one or more natural persons, either directly or indirectly (i.e., in combination with other data) from the dataset?** *If so, please describe how.*

– Yes, our data includes celebrities, or other YouTube-famous people. All of the videos that we use are of publicly available data, following the Terms of Service (`https://www.youtube.com/static?template=terms`) that users agreed to when uploading to YouTube.

Q20. **Does the dataset contain data that might be considered sensitive in any way (e.g., data that reveals racial or ethnic origins, sexual orientations, religious beliefs, political opinions or union memberships, or locations; financial or health data; biometric or genetic data; forms of government identification, such as social security numbers; criminal history)?** *If so, please provide a description.*

– This is highly unlikely, as YouTube removes videos that contain offensive content or do not follow their community guidelines.

Q21. **Any other comments?**

– No.

## Collection Process

Q22. **How was the data associated with each instance acquired?** *Was the data directly observable (e.g., raw text, movie ratings), reported by subjects (e.g., survey responses), or indirectly inferred/derived from other data (e.g., part-of-speech tags, model-based guesses for age or language)? If data was reported by subjects or indirectly inferred/derived from other data, was the data validated/verified? If so, please describe how.*

– See Q7 for an explanation of how the candidate video IDs were chosen. These video IDs were provided by the YT-Temporal-180M dataset providers [17] and collected via the YouTube API. The "video_id" and "URL" are directly observable. The "chapters" are extracted from the YouTube description which is directly observable. The "asr" is obtained by applying the Whisper-Large-V2 model [9] to the directly observable audio from the video. We found this model to provide higher-quality transcriptions compared to the YouTube API on several data samples from VidChapters-7M.

Q23. **What mechanisms or procedures were used to collect the data (e.g., hardware apparatus or sensor, manual human curation, software program, software API)?** *How were these mechanisms or procedures validated?*

– We collected all data using compute resources provided by IDRIS, under the allocation 2023-A0131011670 made by GENCI. The code for querying APIs, extracting ASR, and filtering data are implemented in Python. The code was validated by checking several data samples from VidChapters-7M.

Q24. **If the dataset is a sample from a larger set, what was the sampling strategy?**

– See Q7.

Q25. **Who was involved in data collection process (e.g., students, crowd-workers, contractors) and how were they compensated (e.g., how much were crowd-workers paid)?**

– Our data collection pipeline is fully automatic and does not require any human annotators. YouTube users have uploaded videos whose metadata is a part of VidChapters-7M – we did not directly interact with these users.

Q26. **Over what timeframe was the data collected? Does this timeframe match the creation timeframe of the data associated with the instances (e.g., recent crawl of old news articles)?** *If not, please provide a description of the timeframe.*

– VidChapters-7M contains videos that were uploaded to YouTube between 2005–2022. We collected all data in early 2023, which we used to conduct experiments for our NeurIPS 2023 submission.

Q27. **Were any ethical review processes conducted (e.g., by an institutional review board)?** *If so, please provide a description of these review processes, including the outcomes, as well as a link or other access point to any supporting documentation.*

– We did not conduct a formal ethical review process via institutional review boards. However, as described in Section 3.3 of the main paper and Q16 we employed several filtering mechanisms to tag instances that could be problematic.

Q28. **Does the dataset relate to people?** *If not, you may skip remaining questions in this section.*

– Yes, see Q17.

Q29. **Did you collect the data from the individuals in question directly, or obtain it via third parties or other sources (e.g., websites)?**

– We collected data submitted by YouTube users indirectly through the YouTube API. However, users agree with YouTube's Terms of Service regarding the redistribution of their data by YouTube.

Q30. **Were the individuals in question notified about the data collection?** *If so, please describe (or show with screenshots or other information) how notice was provided, and provide a link or other access point to, or otherwise reproduce, the exact language of the notification itself.*

– No. YouTube users are not required to share their personal contact information (email, phone numbers, etc.). Hence, the only way to notify the authors of VidChapters-7M videos is by commenting on their videos. This is practically difficult to do manually and will be classified as spam and blocked by YouTube if attempted to programmatically write a templated comment to millions of users.

Q31. **Did the individuals in question consent to the collection and use of their data?** *If so, please describe (or show with screenshots or other information) how consent was requested and provided, and provide a link or other access point to, or otherwise reproduce, the exact language to which the individuals consented.*

– Users did not explicitly consent to the use of their data in our dataset. However, by uploading their data on YouTube, they consent that it would appear on the YouTube plaform and will be accessible via the official YouTube API (which we use to collect VidChapters-7M).

Q32. **If consent was obtained, were the consenting individuals provided with a mechanism to revoke their consent in the future or for certain uses?** *If so, please provide a description, as well as a link or other access point to the mechanism (if appropriate).*

– Users have full control over the presence of their data in our dataset. If users wish to revoke their consent, they can delete the underlying YouTube video – it will be automatically removed from VidChapters-7M since we distributed videos as URLs. Moreover, we provide an opt-out request form on our dataset website for anybody to request removal of an individual instance if it is potentially harmful (e.g. NSFW, violates privacy, harmful stereotypes, etc.).

Q33. **Has an analysis of the potential impact of the dataset and its use on data subjects (e.g., a data protection impact analysis) been conducted?** *If so, please provide a description of this analysis, including the outcomes, as well as a link or other access point to any supporting documentation.*

– No.

Q34. **Any other comments?**

– No.

## Preprocessing, Cleaning, and/or Labeling

Q35. **Was any preprocessing/cleaning/labeling of the data done (e.g., discretization or bucketing, tokenization, part-of-speech tagging, SIFT feature extraction, removal of instances, processing of missing values)?** *If so, please provide a description. If not, you may skip the remainder of the questions in this section.*

– We converted chapter timestamps in HH:MM:SS format to seconds. Refer to Section 3.1 of the main paper for more details. We also extracted speech transcripts and visual features (see Section 3.2 of the main paper). Finally, we tagged some instances with a focus on ethical considerations, see Q16 for more details.

Q36. **Was the "raw" data saved in addition to the preprocessed/cleaned/labeled data (e.g., to support unanticipated future uses)?** *If so, please provide a link or other access point to the "raw" data.*

– Yes, the raw descriptions from which chapters are extracted are also released on the dataset website [1].

Q37. **Is the software used to preprocess/clean/label the instances available?** *If so, please provide a link or other access point.*

– Yes, the data preprocessing code is open-sourced and accessible from the dataset website [1].

Q38. **Any other comments?**

– No.

## Uses

Q39. **Has the dataset been used for any tasks already?** *If so, please provide a description.*

– We have used our dataset to train deep neural networks that perform video chapter generation, and that can be transferred to dense video captioning tasks (see Sections 4.1 and 4.4 in the main paper). We also trained models for video chapter generation with ground-truth boundaries and video chapter grounding (see Sections 4.2 and 4.3 in the main paper).

Q40. **Is there a repository that links to any or all papers or systems that use the dataset?** *If so, please provide a link or other access point.*

– We do not maintain such a repository. However, citation trackers like Google Scholar and Semantic Scholar would list all future works that cite our dataset.

Q41. **What (other) tasks could the dataset be used for?**

– We anticipate that the dataset could be used for a variety of video-and-language tasks, such as text-to-video retrieval.

Q42. **Is there anything about the composition of the dataset or the way it was collected and preprocessed/cleaned/labeled that might impact future uses?** *For example, is there anything that a future user might need to know to avoid uses that could result in unfair treatment of individuals or groups (e.g., stereotyping, quality of service issues) or other undesirable harms (e.g., financial harms, legal risks) If so, please provide a description. Is there anything a future user could do to mitigate these undesirable harms?*

– This is very difficult to anticipate. Future users of our dataset should be aware of YouTube's user demographics which might subtly influence the types of videos, languages, and ideas that are present in the dataset. Also, note that our dataset is mainly composed of English videos, hence models trained on this dataset might perform worse on videos in other languages.

Q43. **Are there any tasks for which the dataset should not be used?** *If so, please provide a description.*

– Broadly speaking, our dataset should only be used for non-commercial academic research. Our dataset should not be used for any tasks that involve identifying features related to people (facial recognition, gender, age, ethnicity identification, etc.) or making decisions that impact people (mortgages, job applications, criminal sentences; or moderation decisions about user-uploaded data that could result in bans from a website). Any commercial and for-profit uses of our dataset are restricted – it should not be used to train models that will be deployed in production systems as part of a product offered by businesses or government agencies.

Q44. **Any other comments?**

- No.

## Distribution

Q45. **Will the dataset be distributed to third parties outside of the entity (e.g., company, institution, organization) on behalf of which the dataset was created?** *If so, please provide a description.*

- Yes, our dataset is publicly available.

Q46. **How will the dataset will be distributed (e.g., tarball on website, API, GitHub)** *Does the dataset have a digital object identifier (DOI)?*

- We distribute our dataset as JSON/PICKLE files containing annotations. Users will have to download the videos by themselves by using our data collection code. All uses of VidChapters-7M should cite the paper as the reference.

Q47. **When will the dataset be distributed?**

- The dataset is publicly available as of September 2023.

Q48. **Will the dataset be distributed under a copyright or other intellectual property (IP) license, and/or under applicable terms of use (ToU)?** *If so, please describe this license and/or ToU, and provide a link or other access point to, or otherwise reproduce, any relevant licensing terms or ToU, as well as any fees associated with these restrictions.*

- Uses of our dataset are subject to YouTube API terms (`https://www.youtube.com/static?template=terms`). The data and code are released with an MIT license.

Q49. **Have any third parties imposed IP-based or other restrictions on the data associated with the instances?** *If so, please describe these restrictions, and provide a link or other access point to, or otherwise reproduce, any relevant licensing terms, as well as any fees associated with these restrictions.*

- The videos corresponding to our instances are legally owned by YouTube users. Our dataset users can download them from the URLs we provide in annotation files, but redistributing videos for commercial use is prohibited.

Q50. **Do any export controls or other regulatory restrictions apply to the dataset or to individual instances?** *If so, please describe these restrictions, and provide a link or other access point to, or otherwise reproduce, any supporting documentation.*

- No.

Q51. **Any other comments?**

- No.

## Maintenance

Q52. **Who will be supporting/hosting/maintaining the dataset?**

- The authors will maintain the dataset. The dataset is hosted using Inria servers and Google Drive service. All the information about the dataset, including links to the paper, code, and future announcements will be accessible at the dataset website [1].

Q53. **How can the owner/curator/manager of the dataset be contacted (e.g., email address)?**

- The contact emails of authors are available on the dataset website [1].

Q54. **Is there an erratum?** *If so, please provide a link or other access point.*

- There is no erratum for our initial release. We will version all errata as future releases (Q55) and document them on the dataset website [1].

Q55. **Will the dataset be updated (e.g., to correct labeling errors, add new instances, delete instances)?** *If so, please describe how often, by whom, and how updates will be communicated to users (e.g., mailing list, GitHub)?*

– We will update our dataset once every year and announce it on the dataset website [1]. These future versions would remove instances that were requested to be removed via the opt-out form (Q32).

Q56. **If the dataset relates to people, are there applicable limits on the retention of the data associated with the instances (e.g., were individuals in question told that their data would be retained for a fixed period of time and then deleted)?** *If so, please describe these limits and explain how they will be enforced.*

– Rather than directly distributing videos, we distribute URLs that point to the original videos uploaded by YouTube users. This means that users retain full control of their data – any post deleted from YouTube will be automatically removed from VidChapters-7M (see also Q10, Q14, Q31).

Q57. **Will older versions of the dataset continue to be supported/hosted/maintained?** *If so, please describe how. If not, please describe how its obsolescence will be communicated to users.*

– A new version release of VidChapters-7M will automatically deprecate its previous version. We will only support and maintain the latest version at all times. Deprecated versions will remain accessible on the dataset website for a few weeks, after which they will be removed. We decided to deprecate old versions to ensure that any data that is requested to be removed (Q32) will be no longer accessible in future versions.

Q58. **If others want to extend/augment/build on/contribute to the dataset, is there a mechanism for them to do so?** *If so, please provide a description. Will these contributions be verified? If so, please describe how. If not, why not? Is there a process for communicating/distributing these contributions to other users? If so, please provide a description.*

– Anyone can extend VidChapters-7M by using our data collection code (linked on our website [1]). We are open to accepting extensions via personal communication with contributors. Otherwise, our code and data licenses allow others to create independent derivative works (with proper attribution) as long as they are used for non-commercial academic research.