# OpenReview forum: "VidChapters-7M: Video Chapters at Scale"
_NeurIPS.cc/2023/Track/Datasets_and_Benchmarks — NeurIPS 2023 Datasets and Benchmarks Poster_

### Official Review · Reviewer_37uM · 2023-07-19
**The authors propose a long video caption dataset, and the pretrained model from this dataset show the performance improvement on YouCook2 and ViTT.**

**Rating:** 5
**Confidence:** 4
**Correctness:** Yes.
**Clarity:** Yes.

**Strengths:**

1. Pretraining on long video caption is important and interesting.

**Additional Feedback:**

The details feedback can be found in "Opportunities For Improvement".

**Documentation:**

Yes.

**Ethics:**

No.

**Limitations:**

No.

**Opportunities For Improvement:**

1. The problem of long video caption is not a very new problem. Compared with the existing video caption datasets, VidChapters-7M focus on long video. However, for long video, the probablistic reasoning is more interesting problem for fine-grained video caption. It is better for the authors to construct more fined-grained datasets based on VidChapters-7M.
2. The collected data from the website may be noisy and biased. It is better for authors to add how to de-noise and de-biase these collected data.
3. For the captions with different languages, it is better for authors to provide the multi-lingual video caption baseline and benchmark.

**Relation To Prior Work:**

The authors provide a long video caption datasets.

**Summary And Contributions:**

The authors present VidChapters-7M, which is composed of 817K user-chaptered videos. The authors create the dataset by scraping user-annotated chapters. They propose the benchmark for video paragraph, video caption and grounded video caption. They evaluate their method by pretraining on VidChapters and then transfers to dense video captioning tasks on the YouCook2 and ViTT benchmarks.

---

> ### Author Response · Authors · 2023-08-10
> **Author response to Reviewer 37uM**
>
> We thank Reviewer 37uM for providing interesting feedback.
>
> **Multi-lingual baseline and benchmark:** As suggested, we have conducted new experiments and report video chapter generation results on VidChapters-7M split by languages for both English and German in Tables 1 and 2 of the Supplementary Material. We find that training on VidChapters-7M is beneficial for both languages. Interestingly, pretraining on HowTo100M (which is a dataset in English) improves results on English as well as German. We also observe that the quantitative results in German are lower than in English. Finally, we report results of Vid2Seq with the multi-lingual language model mT5 pretrained on the multi-lingual dataset mC4 (https://arxiv.org/abs/2010.11934). We find that this variant performs a bit worse on English but similar on German compared to the Vid2Seq variant based on T5 pretrained on the C4 corpus.
>
> **Noise and biases:** We agree with the reviewer that noise and biases are important concerns for large-scale datasets. We study biases in Sections 3.3 and 5. To assess the quality of chapter title annotations in our dataset, we inspected a random sample of 100 videos in VidChapters-7M. For each video, we checked if the titles are related to the content of the video chapter and if so which video modalities (ASR, visual or raw audio) they are related to, or if they only refer to the structure of the video (e.g. chapter titles like "step 1", "step 2" etc). Results are presented in the new Table 8 of the main paper, and show that 83% of videos have chapters related to one or multiple modalities of the video, 14% of videos have chapters only referring to the structure of the video, and 3% of videos have chapters unrelated to the video content. We note that despite the noise, our dataset enables pretraining state-of-the-art dense video captioning models, while being smaller than other recent Web video datasets such as HowTo100M (see Tables 6 and 7). Moreover, there are multiple options that could be explored to denoise or debias our large-scale dataset. For instance, one could check that the video frames contain a high enough number of objects as measured by an object detector to maximize their visual interest, or that they are dynamic enough measured by the CLIP feature similarity score between different video frames. The data collection could target specific video categories, languages or genders which are least represented in our dataset. We hope that our provided dataset and analysis will foster research in these directions.
>
> **The problem of long video caption is not a very new problem**: The type of data that we propose (chapters) is new compared to prior open-sourced datasets (see Table 1). Note that Reviewer UiaR underlines that the video chapter generation task we study is “not commonly explored for long videos”, and that both Reviewer Tr45 and Reviewer 4ifN highlight the “practical” application of this task. Moreover, our dataset is much larger than current dense video captioning datasets, and our chapter annotations differ significantly from ASR explored in prior work in the paper as explained in Section 3.3 and shown experimentally in Section 4.4. We would be grateful to Reviewer 37uM to clarify the meaning of the “probabilistic reasoning for long video”. In addition, understanding the high-level structure of long videos is not a solved problem, as highlighted by the ATP paper (https://arxiv.org/abs/2206.01720). We agree with the reviewer that studying fine-grained tasks in long videos is an interesting research direction. However, we note that the tasks we tackle are “challenging” as noted by Reviewer Tr45. Moreover, our VidChapters-7M dataset provides two levels of granularity of annotations with chapters and ASR. We leave the non-trivial construction of more fine-grained datasets at the scale of VidChapters-7M for future work.

---

### Official Review · Reviewer_4ifN · 2023-07-21
**[Review] VidChapters-7M: Video Chapters at Scale**

**Rating:** 7
**Confidence:** 4
**Correctness:** The data collection process makes sense.

**Strengths:**

- Generating video chapters is a practical application of modern media platforms. In addition, the proposed VidChapter-7M can directly improve the performance of the task.

- The definition of three tasks for video chapters is reasonable, while the authors describe and compare their difference from conventional tasks including dense video captioning and scene segmentation.

- The data collection process is well-described for readers to easily understand it.

- The authors have demonstrated the potential of the VidChapter-7M dataset to improve not only video chapter generation performance but also transfer learning for dense video captioning.

- Section 2 provides in-depth discussion to compare the proposed dataset and tasks with previous datasets and video tasks.


**Additional Feedback:**

Please refer to the comments above.

**Clarity:**

The paper is well-written, but some details should be explained. Please refer to the comments above.

**Documentation:**

The authors provide enough information for availability and maintenance of the dataset.

**Limitations:**

The authors have acknowledged the limitations of this study, particularly the biases in the distribution of video categories. However, I would encourage the authors to discuss additional limitations of the collected dataset, such as the accuracy of user-provided annotations.


**Opportunities For Improvement:**

- In Section 3.2, the section on ASR extraction lacks detailed examples. Specifically, it's unclear how the accurate word-level timestamps are generated, as mentioned in Line 145-146. The authors should provide some examples or visualized workflow for exact understanding of readers.

- In the experimental section, the authors should provide the performance results of Vid2Seq or Moment-DETR without using VidChapter-7M to demonstrate the effectiveness of fine-tuning on VidChapter-7M. For example in Table 4, the performance of Vid2Seq without fine-tuning is absent.

- Although the authors consider biases and ethical considerations of collected videos, measuring the quality of collected video can provide useful information. For example, the authors can measure the similarity of chapter texts and video frames to examine the text-video alignments or the aesthetic score of video frames to measure their visual quality.


**Relation To Prior Work:**

The authors clearly discuss about the previous studies.

**Summary And Contributions:**

This study presents the VidChapter-7M dataset, which includes 817K videos and 6.8M user-annotated chapters. The video samples are collected from YouTube using the URLs of YT-Temporal 1B. This study also defines three tasks of video chapter generation, video chapter generation given ground-truth boundaries, and video chapter grounding. Then, the authors have shown that the VidChapter-7M dataset can improve the performance of the defined tasks as well as conventional tasks of dense video captioning.

---

> ### Author Response · Authors · 2023-08-10
> **Author response to Reviewer 4ifN**
>
> We thank Reviewer 4ifN for providing interesting feedback.
>
> **Illustration of the ASR workflow:** We thank the reviewer for the valuable comment. The ASR word-level timestamps are generated following WhisperX [6], which provides a workflow visualization at https://github.com/m-bain/whisperX/blob/main/figures/pipeline.png. We next provide an example of an ASR segment generated with Whisper-V2-XLarge: “Right, we're gonna do the Synthetics Dirty Race. No we're not. In fact, let's change up a bit. We're gonna do the Cotton Fort Dirty Race. On both. This is gonna be interesting. So we're gonna put two t-shirts and two pairs of jeans in the” spoken between start=20.478s and end=50.465. To derive temporally localized sentences from this text, we split it by sentences and leverage the word-level timestamps generated by WhisperX. The first sentence output by this pipeline is “Right, we're gonna do the Synthetics Dirty Race.” spoken between start=20.538s and end=29.26s. We will add representative examples like this one in the final version.
>
> **Effectiveness of finetuning on VidChapters-7M:** As suggested, we added the results of Vid2Seq pretrained on C4 and HowTo100M (without finetuning on VidChapters-7M) for video chapter generation in Tables 2, 3 and 4 and the results of Moment-DETR pretrained on 5.4K narrated videos using the checkpoint from [44] (without finetuning on VidChapters-7M) for video chapter grounding in Table 5. As can be observed, the models finetuned on VidChapters-7M achieve much higher results, which demonstrates the effectiveness of finetuning on VidChapters-7M.
>
> **Measure of the quality of collected video:** To measure the text-video alignment, we compute the CLIP cosine similarity between chapter titles and their corresponding video frames and plot the resulting distribution in Figure 3 of the Supplementary Material (page 14). The average similarity score is 54.6%, and less than 1% of the chapters have a visual-text similarity score below 30%. These statistics demonstrate a good video-text alignment in VidChapters-7M. Moreover, we would be grateful to Reviewer 4ifN to provide an example of a model that can be used to compute aesthetic scores of video frames, and we will be happy to run it and add the results to the paper.
>
> **Accuracy of user-provided annotations:** To assess the quality of chapter title annotations in our dataset, we inspected a random sample of 100 videos in VidChapters-7M. For each video, we checked if the titles are related to the content of the video chapter and if so which video modalities (ASR, visual or raw audio) they are related to, or if they only refer to the structure of the video (e.g. chapter titles like "step 1", "step 2" etc). Results are presented in the new Table 8 of the main paper, and show that 83% of videos have chapters related to one or multiple modalities of the video, 14% of videos have chapters only referring to the structure of the video, and 3% of videos have chapters unrelated to the video content. We note that despite the noise, our dataset enables pretraining state-of-the-art dense video captioning models, while being smaller than other recent Web video datasets such as HowTo100M (see Tables 6 and 7).

---

> > ### Comment · Reviewer_4ifN · 2023-08-11
> > **Rebuttal Acknowledgement**
> >
> > Thanks for the authors' feedback.
> > Since all answers have resolved my concerns, I'm happy to increase my score for acceptance.

---

### Official Review · Reviewer_Tr45 · 2023-07-21
**VidChapters-7M review**

**Rating:** 7
**Confidence:** 4
**Clarity:** Yes, throughout the language and over…

**Strengths:**


This paper presents a novel, large-scale dataset of videos with human-provided chapter titles, and proposes several variants of the video chaptering task.  This is an important and practical problem with immediate applications, and has largely been overlooked.

The proposed tasks appear quite challenging, justifying the need for a large dataset.

The idea of using chapter annotations is intuitively appealing: chapter annotations can be expected to be much more relevant to the video content than noisy ASR transcriptions; they are explicitly constructed to be a high-level description of a video segment.

Transfer results to dense captioning (table 6) are quite impressive.  The VC7M dataset is quite complementary to even a large dataset like HTM.

Experiments show that scale is clearly valuable for pretraining in the dense captioning task.

**Additional Feedback:**

n/a

**Correctness:**

The construction is sound, though as the authors point out, it is limited by the distribution of the underlying dataset YT-Temporal-180M and inherits its overall distribution from that work.  The authors do a reasonable job of presenting the distribution of data in terms of video category, language, etc. and the experiments are thorough and convincing.

**Documentation:**

The paper includes a URL to a webpage with code for download.  Checkpoints are not yet provided but the code appears to be sufficient to repro.

The construction explicitly describes steps taken to identify harmful content and flags these videos in the dataset.  Also included is discussion of biases including language and gender.

Videos were sourced starting from candidates from a large pool used in the construction of YT-Temporal-180M and following a number of filters on things like existence of chapter descriptions in a clear format, which are clearly described in the text.

**Ethics:**

Scraping YouTube data is increasingly being scrutinized.  However, this paper does nothing beyond what other YT-sourced datasets have done previously, and is common practice in the video community.  No new issues have been introduced here and the accompanying web page appears to provide a way to request removal of videos.



**Limitations:**

This work does not immediately suggest negative applications.  The authors suggest surveillance as a possible downstream application, indeed, any applications which might benefit from searching a large corpus of video data could benefit from the approaches and data provided here.


**Opportunities For Improvement:**

When discussing the multimodal nature of the task in 4.1, should probably mention that visual features benefit from large-scale pretraining.  For example, Vid2Seq (Speech+Visual) with only C4 pretraining exhibits a relatively small gain compared to Vid2Seq (Speech), and the task is dominated by the speech signal.  This is mentioned in 4.2, but the comment seems to apply to all settings.

I would like to understand better the value of ASR vs Chapter annotations in applications, e.g. in the transfer experiments.  If it is possible to ablate ASR vs chapter in some settings that would be interesting.  ASR is presumably much denser, but much of the text can be expected to be irrelevant to the visuals in many settings, unlike the Chapter annotations.  It would be interesting to know the relative value of each kind of annotation.

Similar, it would be interesting to have some qualitative examples of the produced captions / chapters.  Do the speech+visual models do better in ways that the numbers don't fully reflect?  How do the VC-pretrained captioning models improve vs. non-VC?



**Relation To Prior Work:**

Yes, a number of current and related works are presented and the present work is compared both in terms of content and scale.  It contains both ASR transcripts and chapter annotations.  The chapter annotations can be expected to be much more relevant to the video content than noisy ASR transcriptions.

**Summary And Contributions:**

The authors identify the lack of a good dataset for the task of video segmentation into chapters, and construct such a dataset by scraping user-annotated chapter markers and text descriptions from ~800K YouTube videos.  Experiments on the new dataset include several settings: chapter generation (with and without temporal boundary GT), and chapter temporal localization from text query.

---

> ### Author Response · Authors · 2023-08-10
> **Author response to Reviewer Tr45**
>
> We thank Reviewer Tr45 for providing interesting feedback.
>
> **Visual features benefit from large-scale pretraining**: We agree with the reviewer that our experiments in multiple settings show that the visual signal is mainly helpful when using large-scale pretraining on HowTo100M. We believe this is because visual cues are dense and pretraining is often required to leverage them effectively [110], while speech is a lower bandwidth modality which requires less data to be effectively used. We will clarify this in the final version.
>
> **ASR vs Chapters in the transfer experiments**: Our original experiments in Tables 6 and 7 show that training on both ASR and chapter annotations largely outperforms training only on ASR annotations for the downstream dense video captioning task (compare rows “C4 + HTM + VC (ASR)” and “C4 + HTM + VC (ASR+Chap.)”). To better understand the value of each kind of annotation, we have added in Table 7 (in red color) new zero-shot results of Vid2Seq trained only on chapter annotations or only on ASR annotations from VidChapters-7M in the visual-only and the visual+speech settings. We find that in the visual-only setting, training on chapter annotations is better than training on ASR annotations. In the visual+speech settings, using either ASR or chapter annotations does not perform well, as ASR has a large domain gap with dense video captions [110], and training only on chapters does not enable the model to learn how to use the input speech modality. However, using both ASR and chapter annotations results in a largely better zero-shot dense video captioning performance, demonstrating the complementary nature of the ASR and chapters annotations.
>
> **Qualitative examples of the predictions**: We present new qualitative results for video chapter generation and dense video captioning in Figures 4 and 5 of the Supplementary Material (pages 14 and 15). Compared with the speech-only model, a key advantage of the speech+visual video chapter generation model is that it can generalize to videos that do not have ASR input, as shown in the first example of Figure 4 of the Supplementary Material. Compared with the visual-only variant, the multi-modal model can also benefit from speech cues, as seen in the second example of Figure 4 of the Supplementary Material. Moreover, we observe that the dense video captioning model pretrained on VidChapters-7M is more accurate and hallucinates less, see Figure 5 of the Supplementary Material.

---

### Official Review · Reviewer_UiaR · 2023-07-21
**A novel video understanding dataset for the community.**

**Rating:** 6
**Confidence:** 5
**Clarity:** Yes, this paper is well-written.

**Strengths:**

1. Exploration of a less explored task: The paper introduces the video chapter generation task, which is not commonly explored for long videos. This is a valuable contribution to the field.
2. Large-scale dataset: The creation of a large-scale dataset with ASR and chapter-level annotations is commendable and can serve as a valuable resource for the research community.
3. Extensive evaluation: The paper evaluates numerous models on the video chapter generation task and two related tasks, providing a comprehensive assessment of their performance.


**Additional Feedback:**

Please refer "Opportunities For Improvement" part.

**Correctness:**

This paper proposes VidChapters-7M, a dataset of 817K user-chaptered videos including 7M chapters in total. The proposed dataset is constructed in a sound way.

**Documentation:**

Yes, there are sufficient details.

**Limitations:**

The authors have adequately addressed the limitations and potential negative societal impact of their work.

**Opportunities For Improvement:**

1. Explanation of task challenges: While the video chapter generation task is not well-explored for long videos, it would be beneficial for the authors to explain the specific challenges involved in this task rather than solely focusing on the differences from existing benchmarks.
2. Annotation quality for chapter title: The manuscript raises concerns about the annotation quality for chapter titles. The short lengths and limited descriptions in chapter titles, as evidenced by Figure 3 in main body and Figure 1&2 in supplement, may raise questions about the quality and informativeness of the annotations.

**Relation To Prior Work:**

Yes

**Summary And Contributions:**

The manuscript introduces a new dataset called VidChapters-7M for video chapter generation, chapter title generation, and video chapter grounding tasks. Several baselines and state-of-the-art video-language models are evaluated, and the results show their potential transferability for dense video captioning on other datasets. The paper's strengths include the exploration of the video chapter generation task for long videos and the collection of a large-scale dataset with ASR and chapter-level annotations.

---

> ### Author Response · Authors · 2023-08-10
> **Author response to Reviewer UiaR**
>
> We thank Reviewer UiaR for providing interesting feedback.
>
> **Explanation of task challenges:** We thank the reviewer for the valuable comment. The video chapter generation task requires processing videos significantly longer than in current datasets, spanning over dozens of minutes (see Table 1). Specifically, temporally segmenting a video into chapters involves temporal understanding as well as precise localization. In addition, since chapters are related to each other, producing chapter descriptions requires understanding the high-level context and the structure of the video (see Figure 1). Furthermore, our experiments highlight the importance of multi-modal understanding of transcribed speech and visual content for this task (see Table 2). Moreover, generating a chapter title involves summarizing a variety of low-level information into a high-level, informative but concise piece of text. Finally, since the videos are diverse, this task requires generalizing to videos with various types (such as howto, education, blogs, entertainment, sports, see Figure 3d for details) with different structures. We will clarify these points in the final version.
>
> **Annotation quality for chapter title:** We note that the chapter titles are manually written and uploaded by real users. The short length of chapter titles can be useful in practice, as it helps viewers to quickly identify the content of the video chapter. However, the reviewer is correct in noticing that sometimes  chapter titles are not informative about the content of the video at the corresponding timestamps. To assess the quality of chapter title annotations in our dataset, we inspected a random sample of 100 videos in VidChapters-7M. For each video, we checked if the titles are related to the content of the video chapter and if so which video modalities (ASR, visual or raw audio) they are related to, or if they only refer to the structure of the video (e.g. chapter titles like "step 1", "step 2" etc). Results are presented in the new Table 8 of the main paper, and show that 83% of videos have chapters related to one or multiple modalities of the video, 14% of videos have chapters only referring to the structure of the video, and 3% of videos have chapters unrelated to the video content. We note that despite the noise, our dataset enables pretraining state-of-the-art dense video captioning models, while being smaller than other recent Web video datasets such as HowTo100M (see Tables 6 and 7).

---

### Author Response · Authors · 2023-08-10
**Author response - Overview**

We thank the reviewers for their helpful comments. We appreciate their enthusiasm for our proposed large-scale dataset with user-annotated chapters (UiaR, Tr45) and for our exploration of the video chapter generation task (UiaR, Tr45, 4ifN) that has practical applications (Tr45, 4ifN). Reviewers also appreciated our extensive evaluation (UiaR), and our transfer results to dense video captioning (Tr45, 4ifN) which show the importance of the scale of our dataset (Tr45). Moreover, they found that our paper is clear (UiaR, Tr45, 4ifN, 37uM) and relates to prior work (UiaR, Tr45, 4ifN, 37uM), that our dataset construction is sound (UiaR, Tr45) and well-described (4ifN), and that the documentation of our dataset is sufficient (UiaR, Tr45, 4ifN, 37uM).

We provide detailed answers to the comments and questions from each reviewer in the different author responses and will modify our paper accordingly. Note that we have already included new material (tables and figures) in the paper in red color to complement our responses. Discussion of these materials will be added to the paper.

---

### Decision · Program_Chairs · 2023-09-22

**Decision:**

Accept (Poster)

**Comment:**

This paper considers to solve the problem of segmenting untrimmed videos into chapters. To this end, authors provided a dataset and evaluate three related tasks.
Thanks reviewers to provide insightful comments and, to some extent, and we almost reach the consistency that this paper has made some substantial contribution to the communities. We have four reviewers to return the comments and the ratings are: 6 7 7 and 5. Authors provided the response to the comments from reviewers and two reviewers express their thankfulness.
Based on the reviewer's comments and author's responses, I recommend acceptance of this paper.